



# Third Revision of the Global Surface Seawater Dimethyl Sulfide Climatology (DMS-Rev3)

Shrivardhan Hulswar[1], Rafel Simo[2], Martí Galí[2,3], Thomas G. Bell[4], Arancha Lana[5], Swaleha Inamdar[1,6], Paul R. Halloran[7], George Manville[7] and Anoop S. Mahajan[1*]

5   [1]Indian Institute of Tropical Meteorology, Ministry of Earth Sciences, Pune, India

[2] Institut de Ciències del Mar (CSIC), Barcelona, Catalonia, Spain

[3] Barcelona Supercomputing Center (BSC)

[4] Plymouth Marine Laboratory (PML), Plymouth, UK

[5] Institut Mediterrani d'Estudis Avançats (IMEDEA, UIB-CSIC), Esporles, Balearic Islands,

10   Spain

[6] Institute of Environment and Sustainable Development, Banaras Hindu University,

Varanasi, India

[7]College of Life and Environmental Sciences, University of Exeter, Exeter, UK

*corresponding author: Anoop Sharad Mahajan (anoop@tropmet.res.in)



**Key points:**

- The sea-surface DMS concentration climatology was updated using an upgraded processing algorithm and the inclusion of new data.

- Usage of monthly dynamic biogeochemical province boundaries and DMS variability length scales reduces the patchiness observed in surface mean concentrations seen in the older climatologies.

- DMS-Rev3 estimates the global annual mean at 1.87 nM (2.35 nM without a sea-ice mask), approximately ~4% lower than the last DMS climatology (~4% lower without considering sea-ice), with much larger regional differences.





## Abstract

This paper presents an updated estimation of the bottom-up global surface seawater dimethyl sulfide (DMS) climatology. This update, called DMS-Rev3, is the third of its kind and includes five significant changes from the last climatology, 'L11' (Lana et al., 2011) that was released about a decade ago. The first change is the inclusion of new observations that have become available over the last decade, creating a database of 872,427 observations leading to a ~18-fold increase in raw data as compared to the last estimation The second is significant improvements in data handling, processing, and filtering, to avoid biases due to different observation frequencies which results from different measurement techniques. Thirdly, we incorporate the dynamic seasonal changes observed in the geographic boundaries of the ocean biogeochemical provinces. The fourth change involves the refinement of the interpolation algorithm used to fill in the missing data. And finally, an upgraded smoothing algorithm based on observed DMS variability length scales (VLS) helps to reproduce a more realistic distribution of the DMS concentration data. The results show that DMS-Rev3 estimates the global annual mean DMS concentration to be ~1.87 nM (2.35 nM without a sea-ice mask), i.e., about 4% lower than the previous bottom-up 'L11' climatology. However, significant regional differences of more than 100% as compared to L11 are observed. The global sea to air flux of DMS is estimated at ~27 TgS yr$^{-1}$ which is about 4% lower than L11, although, like the DMS distribution, large regional differences were observed. The largest changes are observed in high concentration regions such as the polar oceans, although oceanic regions that were under-sampled in the past also show large differences between revisions of the climatology. Finally, DMS-Rev3 reduces the previously observed patchiness in high productivity regions.

**Plain Language Summary**

The third climatological estimation of sea-surface DMS concentrations based on in-situ measurements was created. The update includes a much larger input dataset and includes improvements in the data unification, filtering, and smoothing algorithm. The DMS-Rev3 climatology provides more realistic monthly estimates of DMS and shows significant regional differences compared to the past climatologies.

**Keywords:** Dimethyl Sulfide (DMS); Climatology; Sea-air Emissions;



## 1. Introduction

Dimethyl sulfide (DMS) is a volatile compound found in the global oceans and its biogeochemical cycle plays an important role in the Earth's climate system (Andreae and Crutzen, 1997; Charlson et al., 1987). It is primarily a by-product of phytoplankton growth and marine microbial food web interactions (Simó, 2001). DMS is produced by the breakdown of the phytoplankton intracellular metabolite dimethylsulfoniopropionate (DMSP) either in the algal cell or through microbial catabolism of the DMSP released due to physiological stress or mortality (Kiene et al., 2000; Stefels et al., 2007). This produced DMS is either oxidized by photochemical reactions or metabolized by bacteria (Toole et al., 2003), leaving a small portion that is released into the atmosphere as gaseous DMS (Galí and Simó, 2015; Simó, 2001). The DMS emitted from the surface ocean is responsible for up to 70% of the natural sulfur emissions into the global atmosphere (Andreae, 1982; Carpenter et al., 2012). Oxidation of DMS takes place in the atmosphere and yields sulfuric and methanesulfonic acids, which eventually lead to the formation of sulfate aerosols that can grow to act as cloud condensation nuclei (CCN) (Andreae, 2018; Pazmiño et al., 2005). New CCN can make clouds brighter, thus establishing a feedback loop between phytoplankton and cloud albedo popularly known as the CLAW hypothesis (Charlson et al., 1987). While some studies based on large-scale observations of ocean surface DMS provided partial support for the CLAW hypothesis (Vallina and Simó, 2007), other studies based on model sensitivity analysis of the hypothesis have challenged it (Quinn and Bates, 2011; Woodhouse et al., 2010, 2013). However, even if the feedback loop is not as strong as previously envisaged, DMS emissions contribute towards a large fraction of aerosols in the remote oceanic environment (Quinn et al., 2017) and its emission needs to be quantified accurately to improve our understanding of climate sensitivity, the current climate (Carslaw et al., 2013) and improve the accuracy of future projections (Wang et al., 2021).

Acknowledging the significance of oceanic DMS, the scientific community has striven to reproduce an accurate representation of the global atmospheric and seawater DMS concentrations, and more importantly the ocean-atmosphere flux on a global scale. To compute the ocean-atmosphere flux, two basic methods have been adopted. The first method, described here as the top-down approach method, relies on the dependency of DMS on various physical, chemical, and biological parameters that correlate with the variability of DMS, e.g., chlorophyll-*a*, photosynthetically active radiation (PAR), nutrients, etc. This method creates a parameterization-based seawater DMS inventory, which when fed with input fields and used in combination with an air-sea exchange parameterization results in the ocean-atmosphere DMS flux (Belviso et al., 2004; Bopp et al., 2004; Galí et al., 2018; Simó and Dachs, 2002; Vallina and Simó, 2007). These approaches provide statistical relationships needed to understand the mechanisms of the biogeochemical cycle of DMS and incorporating its formation and removal from the surface ocean. These seawater DMS parameterizations produced in the past, help reproduce preliminary or localized DMS concentration fields but have shortcomings when applied on a global scale. For example, the SD02 algorithm (Simó and Dachs, 2002) was able to estimate values accurately only in the tropical and temperate latitudes but underestimated DMS in low chlorophyll areas and along the Antarctic coast. VS07 (Vallina and Simó, 2007) was unable to reproduce the DMS-irradiance relationship that depended on phytoplankton biomass, leading to overestimations or underestimations outside the subtropical region. Another method applies a two-step approach, first computing the DMSP concentrations (Galí et al., 2015) and using them to calculate the DMS concentrations utilizing satellite measured proxies (Galí et al., 2018). However, this satellite-based seawater DMS computation (DMS$_{SAT}$) suffers from a negative bias in the Antarctic coastal region during the productive season. DMS$_{SAT}$ tends to underestimate observations by around 50% in some regions of the Southern Ocean (Galí et al., 2018). A recent parameterization-based approach

used an artificial neural network (ANN) to extrapolate seawater DMS observations into a global climatology (Wang et al., 2020). This approach using linear regressions showed that on a global scale, mixed layer depth (MLD - explaining ~9% of variance) and solar radiation (explaining ~7% of variance) are the strongest predictors of DMS. The ANN climatology captured 66% of the raw data variance in the database. This approach however does not give

much scientific insight into the relationships between biological and physical parameters and processes controlling the DMS concentrations. The concentrations are also under-estimated in the higher latitudes and the episodic occurrence of higher DMS concentrations is also poorly predicted (Bell et al., 2021). These parameterizations (diagnostic models) can however be useful to provide predictions using satellite/model proxies, allowing 'real time' predictions and

interannual variability studies, although there is a clear need to improve them.

The second, a more widely-used method is the bottom-up approach, which relies on the Global Surface Seawater DMS Database (GSSDD) (NOAA-PMEL, 2020), which consists of data contributed by research groups from all over the world. This database was established to consolidate all the surface seawater DMS measurements in a single comprehensive dataset.

This data is then modelled using a combination of smoothing and interpolation techniques to create a gridded seawater DMS concentration climatology, which is then converted into a flux as mentioned above (Kettle et al., 1999; Lana et al., 2011). The first attempt in creating the bottom-up climatology was made by Kettle et al. (Kettle et al., 1999). They used the then available dataset (15,617 observations across the global ocean) to create a global DMS

climatology (hereafter called the K99). A decade later, this climatology was updated by Lana et al. (Lana et al., 2011), using an updated DMS database (47,313 observations) and included some minor changes in the computation algorithm (hereafter called the L11 climatology). Significant differences were observed between the two climatologies: globally, the L11 estimated emission of DMS was 28 (17.6 – 34.4) Tg S year[-1], about 17% higher than the

estimate calculated using K99. Regionally, large differences were observed, for example in the

Indian Ocean, where the L11 predicted higher values, and in the Southern Ocean, where large

longitudinal differences were reported. At present, the L11 (Lana et al., 2011) bottom-up

climatology is considered as the primary reference product for global DMS seawater

concentrations and is used as an input in numerous atmospheric chemistry models (Mahajan et

al., 2015) and climate and Earth system models.

However, over time, significant shortcomings in L11 have been identified. First, it uses a single

threshold for data selection using the 99.9 percentile of the data to remove any extreme values.

This causes the relatively higher values in the open ocean to remain if they are under the

threshold. These might represent a highly productive bloom in the region and show

anomalously high concentrations. Such abnormal values cause regional patches of higher

concentrations which are seen in the L11 climatology. The L11 climatology also uses static

biogeochemical province boundaries for global data segregation, which does not capture

variability in the biogeochemical properties affecting DMS production, especially on seasonal

scales. Since the number of available observations of sea-surface DMS concentrations has

increased, the data distribution over the oceans, especially in under-sampled regions like the

remote oligotrophic oceans and the Southern Ocean has improved (Figure 1). These regions

had little data in the L11 climatology and were hence heavily reliant on interpolations (Tesdal

et al., 2016). The newer data from these remote regions will help to make realistic estimates of

concentrations, hence reducing dependence on the interpolated estimates.

Here we present 'DMS-Rev3', the third revision of the bottom-up DMS climatology, wherein

we have amended the algorithm keeping in mind the shortcomings of K99 and L11. We updated

the DMS database using the latest additions in the GSSDD along with other published data that

are not currently in the database to reconstruct the monthly, seasonal, and annual DMS

climatologies. Comparison with the L11 climatology demonstrates that the DMS-Rev3

addresses some of the concerns with previous climatologies. Furthermore, shortcomings of the latest revision are discussed, along with the identification of gaps that need to be addressed in subsequent versions.

## 2. Methodology

### 2.1. Data consolidation and clean-up

Since the last bottom-up DMS climatology (L11) was published about a decade ago by Lana et al. (Lana et al., 2011), the GSSDD database (NOAA-PMEL, 2020) has been continuously updated with new observations and now consists of a total of 87,801 data points. This is a significant increase in the number of data points (an increase of ~85.6%) compared to the number of data points used in the L11 climatology (47,313 – blue circles in Figure 1). Most of

these observations were made using the gas chromatography (GC) technique and have a temporal resolution ranging from 10 minutes to a data point every week for individual campaigns. Over the last decade, newer techniques that use high-resolution mass spectrometry have become more common and led to a drastic increase in the number of available data points. We added the raw high temporal resolution data (frequency as high as ~1 s) from published

campaigns, which make use of these new techniques (Behrenfeld et al., 2019; Jarníková et al., 2016; Royer et al., 2015; Zavarsky et al., 2018b). On the addition of these high-resolution data, the consolidated raw dataset included 872,427 data points (~1744% increase over the number of data points in the L11 climatology).

The first step after consolidating all the available data was to ensure data quality. To do this,

the values with incorrect location data were removed (in all about 61 data points had no location data and were hence removed). Unfortunately, as explained by Lana et al. (Lana et al., 2011), there is no robust criteria or accepted method for the selection or elimination of historical data. However, intercomparison studies show that the data is expected to be within a range of ±25%

(Bell et al., 2012; Swan et al., 2014). Hence, to avoid the undesirable effects that potentially

erroneous and extreme values might produce during the objective analysis, a two-step filtering

was conducted. The lower limit for the observations was set to 0.001 nM, thus filtering out

incorrectly reported negative values or values below the detection limit of the instruments

(typical detection limits are higher than the set lower limit) resulting in 872,344 data points.

The K99 and L11 climatologies also applied a 99.9 percentile upper threshold filter to remove

the extremely high values which could be due to erroneous observations or a result of

preferential sampling in phytoplankton blooms that could bias the climatology, as suggested

by Galí et al. (Galí et al., 2018). However, the range for DMS values is expected to be different

in different biogeochemical regions. For example, peak values in the open ocean oligotrophic

regions can be up to two orders of magnitude lower than in highly productive coastal

environments. Hence, a single threshold is not applicable for the global dataset. Instead, several

thresholds need to be calculated for data segregated according to the biogeochemical properties

that can affect DMS production. This was done by first sorting the data according to the updated

dynamic Longhurst provinces (detailed in section 2.3). Then the 99.9 percentile threshold was

calculated for every province. This helped identify and remove extreme values in each

province, thereby reducing the number of data points to 871,455 (~0.001 % were rejected).

Details of the DMS concentration percentile thresholds for each province, along with the

amount of data before and after applying the filters is given in Table 1. The range of thresholds

for provinces that had sufficient data (section 2.4) was between 5.5 nM in the northwest

Atlantic to 344.1 nM close to Antarctica (Table 1). By comparison, L11 used a single threshold

of 148 nM for the global oceans, which filtered out the higher values only in coastal and highly

productive environments.

### 2.2. Data unification (spatial and temporal)



The consolidated data contains observations made using vastly different measurement techniques. Until 2000, the GC based instruments were widely used for the measurement of DMS, but later, high-frequency instruments based on high-resolution mass spectrometry (such as the Chemical Ionization Mass Spectrometer (CIMS) and the Membrane Inlet Mass Spectrometer (MIMS)) started to become more common. This resulted in data with different spatial and temporal resolutions. We analyzed the sampling frequency of the observations (Figure 2a) to identify the most common sampling frequency. It was found to be greater than or equal to 24 hours when all the campaigns were considered. However, this is mostly because older campaigns reported data once a day or less frequently. All more recent campaigns have a much higher sampling frequency, at times as frequent as a data point every second. The high-resolution data collected using the mass spectrometry techniques contributed up to ~59% of the raw global database. However, the high-resolution data represents only 9% of the campaigns (Figure 2b and 2c). The climatology, which is based on observations, can thus become biased towards the data procured using high-resolution instruments. Hence, to avoid this bias and standardize the interpolation field to facilitate further analysis, we binned all the high-resolution data to a frequency of 1 hour. This helps match the high temporal resolution data with the rest of the historical data and results in a unified dataset of 48,567 data points for DMS-Rev3, reducing bias towards a particular measurement technique or campaign while ensuring that enough data points are available in all the provinces for further analysis. Although keeping a higher temporal resolution would result in more data points, this calculation also helps reduce the disparity in the spatial resolution of the dataset, with the average distance between two consecutive data points for a single campaign coming to about 0.2° (assuming a constant ship speed of 10 knots). In certain regions of the oceans, where the data availability is higher, a high-resolution climatology of up to 0.2° spatial resolution could be attempted, but the majority of the world's oceans are under-sampled or moderately sampled and hence a










coarser resolution results in a more realistic climatology. Additionally, since most of the

climate models also work on 1° spatial resolution or lower, this dataset was used to create a 1°

climatology like that produced earlier in K99 and L11.

**2.3. A first guess monthly climatology according to dynamic Longhurst provinces**

Since the data was not uniformly distributed across the oceans, the strategy adopted by the

earlier K99 and L11 climatologies was to segregate the data in different provinces based on

their biogeochemical properties  (chlorophyll-a concentrations, nitrate concentration, salinity,

etc.) as defined by Longhurst et al. (2007) (Figure S1(a)). Computing mean values across these

provinces helped in creating a 'first guess' global distribution, which is the first step towards a

climatology at a coarse resolution. However, satellite images for the biogeochemical

parameters reveal that these features are highly dynamic in terms of geographical extents

(Devred et al., 2007; Oliver and Irwin, 2008; Reygondeau et al., 2013). Hence, using a static

province approach for the whole year (as in K99 and L11), though practical, has an inherent

drawback of not accounting for the spatial/temporal changes in the biogeochemistry. This

affects the estimations of DMS, especially along the borders of the provinces where static

boundaries were used in K99 and L11. To address this, we make use of the dynamic Longhurst

provinces based on the work of Reygondeau et al. (Reygondeau et al., 2013) who defined the

dynamic boundaries based on satellite data between 1997–2007 (Figure S1 (b)). The monthly

data were segregated according to the changing geographical extents for all the provinces and

the means of these separated data were used for creating the 'first guess' fields (Figure S2).

The advantage of using dynamic biogeochemical provinces is that they helped resolve the

environmental, biogeochemical dynamics better on a regional scale, especially in different

seasons. The difference due to the inclusion of the seasonal variation in the province boundaries

is discussed further in Section 3.2.

DMS-Rev3 also provides an option to create 'first guess' fields using medians instead of means across the provinces. Median is not affected by dataset skew due to a few larger values. The median will also minimize the effect of the blooms that drive high DMS emissions. This

approach thus helps re-create background values better than using means but can lead to an overall underestimation in regions where blooms are more common, but observations have not been made frequently enough to capture them. Keeping this in mind, we used the province means for the calculation of the climatology, although the values using medians are also reported.

**2.4. Data substitution, merging and interpolation.**

The number of data points available per province differs greatly depending on the sampling carried out in that region during a particular month. Since these are in situ observations, some regions are adequately sampled, some are moderately sampled, while some are rarely or never sampled due to physical constraints like accessibility, remoteness etc. This results in an uneven

distribution of data in the different provinces with respect to space and time. After segregation, some provinces had data for all 12 months of the year, while others had as few as 1-2 months of data (Figure 3; Table 1). Hence, there was a need to fill the monthly gaps in some of the provinces with data from an appropriate 'donor' province to provide sufficient data for a valid interpolation for the annual variation in each province. Across all the provinces, it was seen

that for provinces with only 1-2 months with data, substitution from (or merging with, in case the receiving province had no data at all) a single province would result in 5 months with data, providing enough data for interpolation. Hence, if a province had data for less than 5 months, it was selected for substitution/merging, thus all the 'receiver' provinces had at least 5 months with data after substitution/merging. The data from the donor province were normalized by

scaling up or down based on the ratio of concentrations of the donor and receiver provinces for the common months. In case the receiving province had no data at all, the data was substituted

from the donor province. Details of the biogeochemical provinces that were substituted or merged are given in Table 1.

In all, 14 provinces that needed substitution or merging, i.e., provinces with less than 5 months

that contained data, were identified. The provinces off the eastern coast of Australia (TASM and the ARCH) were inter-substituted between each other. The coastal province AUSE was used to fill the northern coastal province SUND. NEWZ, although south of New Zealand, was also updated with the data from AUSE due to biogeochemical similarities with AUSE. The data from AUSW, AUSE and SUND were merged to complete AUSW. NPSW and KURO in

the northern Pacific were merged with NPSE. In equatorial regions, the coastal province on the east American coast, CAMR was substituted by CCAL following K99 and L11. The province FKLD received data from the adjacent coastal province BRAZ. The coastal Atlantic provinces GUIN and GUIA were substituted by the adjacent provinces ETRA and WTRA, respectively, following L11. Due to the usage of dynamic provinces, the province GUIN is present only in

November, December, January, and June. The Canary coastal current seems to highly influence the region occupied by GUIN province converting it to CNRY for the rest of the months. The Indian Ocean coastal provinces INDE, REDS, and EAFR are under-sampled despite the importance for the highly populated South Asian region's respective countries. Data from the ARAB province is merged in the INDE and REDS provinces, while EAFR is merged with data

from ISSG.

 The substitution or merging was done to ensure that the dependence on interpolation to estimate the values for the remaining months was as low as possible. However, even after this step, it was obvious that the data required some amount of interpolation as only two provinces had data for all 12 months (NATR and NASTE), while other provinces had data between 11 to

5 months (Figure 3). Hence, to obtain monthly data for each province to estimate the annual trend,  it was interpolated to estimate the missing monthly data following the recommendations



of Lana et al., (2011). The above ensured that annual trends were created for every province (Figure 4).

### 2.5. Incorporation of observed VLS for smoothing.

The 'first guess' database has a uniform distribution within a province as it is the mean value in the province. It also has a sharp and unrealistic transition at the boundaries. Hence, to create a realistic distribution of DMS values at the boundaries, a Shuman filter, which is an unweighted 11-point smoothing filter (Shuman, 1957) was employed. On top of these smoothed first guess fields, a 1° x 1° spatially binned DMS concentration field was

superimposed. This replicates the differences in the DMS concentrations within the individual biogeochemical provinces. After the superimposition, a Barnes filter (Barnes, 1964), which is a convergent weighted-average interpolation scheme where the radius of influence (ROI) is used as a 'weight' was applied. The ROI used by K99 and L11 was 555 km (~5°, close to the Rossby radius in the tropics). The main aim of using such a filter is to ensure that the DMS

variability is captured in the climatology, but this also causes patchiness in the resultant climatology. To check the sensitivity of the algorithm to the ROI value, different fixed values of ROI: 555 km, 100 km, 75 km, 50 km, 25 km, 10 km, and 7.5 km were tested. On a global scale, each ROI resulted in a different global mean, but once the ROI dropped below 25 km, the mean value stabilized at ~2.44 nM (Figure S3). Although the global mean did not change

by much, large regional differences were observed, with smaller ROI values showing less patchiness in the resultant climatology, indicating that choosing the right ROI is crucial for an accurate estimation of the DMS distribution.

The L11 climatology used the 555 km value since little information was available on DMS variability length scales (VLS) in the oceans, which is the distance over which one would

expect the DMS concentration to significantly change as we move over the ocean surface.

However, more recent studies based on high-frequency measurements (Asher et al., 2011; Royer et al., 2015) have been able to quantify the variability in DMS and show that it occurs at a much smaller scale in the order of 15 to 50 km. VLS observations are available in 11 biogeochemical provinces, which were then used to estimate the VLS in other provinces to

create a DMS VLS distribution map (Figure 5). Since the VLS also shows a dependence on the biogeochemistry, the monthly geographical variability in VLS was estimated following the dynamic province boundaries, after applying a Shuman filter as detailed above (Figure 5). This VLS distribution was then used for computing the convergent weighted-average interpolation for DMS using the Barnes filter. This approach is consistent with results from a comprehensive

and objective analysis of seawater DMS VLS observations from around the world (Figure S4). The difference in the resultant DMS climatology due to the inclusion of the VLS information is discussed further in Section 3.3.

### 2.6. Using sea-ice cover masks

In addition to the oceanic sources of DMS, the presence of sea-ice affects the emissions of

DMS. The contribution of sea-ice to the total DMS production during the melt period was simulated by Hayashida et al (Hayashida et al., 2017) and showed episodic spikes of up to 8 $\mu$mol m$^{-2}$ d$^{-1}$. However, the exact extent of this contribution is not known due to the scarcity of field measurements. Further model simulations highlighted the importance of addressing the sea-ice ecosystem separately for better DMS flux estimates. The K99 and L11 DMS

climatologies applied a sea-ice filter to mask out any emissions from areas covered with sea-ice, which most likely underestimates the total DMS flux.

Considering that large portions of the polar waters can be under sea-ice at different times during the year, it is necessary to apply a sea-ice filter to modulate emissions in these regions. This can be done in two ways: i) before creating the DMS climatology to filter out data that are

'under' sea-ice regions; or ii) after the climatology is created as a mask. A sea-ice monthly

climatology (Figure S5) was created using data obtained from the Defense Meteorological

Satellite Program (DMSP) F13 Special Sensor Microwave/Imager (SSM/I) and Scanning

Multichannel Microwave Radiometer (SMMR). The DMS-Rev3 code replaces the data with

zeros when considered to be under sea-ice, where sea-ice is less than or equal to the set

threshold. For example, the Antarctic coastal region is a DMS hotspot during the southern

hemisphere summer and is a sea-ice covered region during the winter with no DMS emissions.

An annual average ignoring these regions during winter would result in a hotspot with the

annual average biased towards the summer values.

A sea-ice threshold cutoff value of 50% was used (which can be changed in the code). If the

filter was applied after data unification, but before the creation of the climatology, the mask

removed only 16 data points (87 data points for 30% sea-ice cover threshold) from the 48,567

datapoints that are deemed to be under the sea-ice. This does not lead to a significant change

in the overall calculations. However, if the mask is applied after the creation of the climatology,

large portions from the polar provinces come under the mask and lead to changes in the monthly

and the annual global mean values (detailed in Section 3.4).

### 2.7. Sea-air DMS flux estimations

The sea-air DMS flux was estimated using the output of DMS-Rev3 to understand the impact

of the new climatology. The estimations were carried out following the same procedure as Lana

et al (Lana et al., 2011) which used the Nightingale et al (2000) parameterization. This was

done for allowing a direct comparison with the previous climatology. This parameterization is

based on the DMS gas transfer velocity utilizing the wind speed at 10 m and the climatological

sea surface temperature. The inputs for calculating the flux were for the same period as that

used for L11 (1978-2008) and were obtained from the NCEP/NCAR reanalysis project

(http://www.esrl. noaa.gov/). This was necessary for a one-to-one comparison of DMS-Rev3

and L11 flux estimations.

### 2.8. Uncertainties in the climatology

Estimating the uncertainty in the climatology is difficult due to the variety of methodologies and the lack of reported observational uncertainties for most of the campaigns. The uncertainties in the DMS climatology were hence estimated from the point of data unification

(Section 2.2) and will therefore be an underestimate of the total uncertainty in the final climatology. First, the standard deviations were computed for the hourly binned observations. This standard deviation was further propagated while spatially averaging the data when creating the 1°x 1° bins. An additional standard deviation was computed for the first guess fields, which use the means across the biogeochemical provinces (Section 2.3). These standard

deviations from the hourly binned data, pixel binned data and province binned data were then propagated through the calculation of pooled standard deviations. The uncertainty introduced due to the smoothing using the Shuman and Barnes filters (Section 2.5) is difficult to estimate and hence the same smoothing filters were also applied to the standard deviations to create a monthly and annual uncertainty database corresponding to the DMS concentrations. While

these uncertainties are not a complete estimation of the errors, they provide an estimate of the range of DMS concentrations in the climatology. The calculated global monthly and annual standard deviations are shown in Figure S6.





## 3. Results and discussion

### 3.1. Features in DMS-Rev3

#### 3.1.1. Global

Figure 6 shows the monthly and annual climatological distribution of surface seawater DMS

concentrations as estimated by the DMS-Rev3 algorithm after the application of a 50% sea-ice

cover mask (the distribution without the application of the sea-ice mask is shown in Figure S7).

For the annual mean climatology, the DMS concentrations range between 0.1 to 5 nM, of which

~83% of the grid points contain DMS concentrations below 3 nM (Figure S8). About 10% of

those points fall under sea-ice (masked as 0 nM). When globally averaged across all the oceans,

DMS-Rev3 estimates the annual mean at 1.87 nM (2.34 nM with no sea-ice mask – henceforth

nM* indicates values without a sea-ice mask). If province medians are considered instead of

the province means, the global annual average concentration reduces to 1.46 nM (1.72nM*),

about ~25% lower than the L11 climatology (~11% lower than DMS-Rev3 calculated using

means). The highest global averages are observed in November, December, January, and

February with average values of 1.99 nM (2.82 nM*), 2.57 nM (3.44 nM*), 2.76 nM (3.57

nM*) and 2.06 nM (2.68 nM*), respectively (Figure S9). The higher values in these months

are mainly due to large concentrations in the southern hemisphere with coastal Antarctica

dominating the peak values. The global mean annual cycle shows a clear peak in December

and January, followed by the first minimum in April ~1.52 nM (~1.82 nM*). The values then

show a modest increase through the months of May-August (with values around ~1.7 nM) due

to the northern hemispheric summer. This can be attributed to the higher DMS values in the

northern high latitudes. September shows the second minimum lowest mean DMS values of

1.40 nM (1.56 nM*). The region below 60°S has the highest average concentrations, with large

peaks seen in the high productivity regions such as the Weddell Sea and the Ross Sea. Outside

this polar environment, higher values are seen on the southeast coast of Africa and the east

coast of South America. Elevated concentrations are also observed in the Mediterranean Sea

and the Arctic Ocean close to Norway and Greenland. The waters close to Alaska and

California also show higher than the global mean concentrations.

A clear regional annual cycle is observed in most locations, with the Southern and Northern

hemispheres peaking during their respective summers. The range of values also differ

according to the months, but higher values of as much as ~8 to 14.7 nM (10-15 nM*) are seen

in the Antarctic coastal province (APLR), especially in November, December, and January.

However, even during these months, more than ~60% (83% in November) of the grid points

contain concentrations below 3 nM (Figure S8). The final output for individual provinces is

shown in Figure S10.

### 3.1.2. Polar

Regionally, the highest values were found in the polar biomes during their respective summers

(60° to 90° both in the northern and southern hemispheres). APLR showed the highest levels

of DMS with peak monthly averaged concentrations of 12.84 nM (13.35 nM*) and 14.71 nM

(15.14 nM*) in December and January (Figure S10). The DMS concentration was 0.57 nM in

September. However, since this region is under the ice during this season, this value

corresponds to the concentration limited to a smaller region to the north of the Weddell Sea,

which stays exposed and out of ice cover (Figure 6). The concentrations increase through spring

and summer until January, after which the values decrease to April which has an average of

1.34 nM (1.4 nM*). Over these months, since most of the APLR province is under sea-ice, it

does not make a large contribution to the global monthly mean DMS. Similarly, most of the

polar province of BPLR also gets masked by sea-ice during the northern winter. The peak DMS

concentration was 3.86 nM (2.32 nM*) during May (Figure S10). In the North Atlantic region,

phytoplankton blooms dominate the Labrador and the Grand Banks coastal regions (ARCT

province) in spring and summer (Friedland et al., 2016), driving high DMS concentrations. A
similar peak is observed in the SARC province with elevated DMS values during the same
seasons (Figure S10).

### 3.1.3. Extra-tropics

The southern extra-tropical region gets split into three sectors: the Indian Ocean sector, the
Atlantic Ocean sector, and the Pacific Ocean sector.  The northern extra-tropical region has
two sectors: the Pacific Ocean and the Atlantic Ocean.

In the southern hemisphere, in the Indian Ocean extra-tropical region, the monthly variation
observed in the polar province APLR is also noticed in the adjacent province ANTA, which
shows peak monthly mean concentrations of 7.77 nM (8.33 nM*) in December and 7.54 nM
(7.68 nM*) in January (Figure S10). The adjacent provinces SANT and SSTC form a part of
all the sectors mentioned above, although divided by the South American and the African
continents. These provinces follow a similar annual variability as the APLR and ANTA
provinces, although the peak values are lower (Figure S10). These concentrations reflect the
summer-time productivity of the Southern Ocean during the summer. The province ISSG in
the Indian Ocean sector, spread over the tropics and extra-tropics, follows a similar seasonal
trend, although being further away from the polar region has much lower month-to-month
variations. The highest concentrations are observed in the months of November-December-
January. During the rest of the year, the values are lower and stay close to ~2 nM, with a slight
increase in June to 2.58 nM (Figure S10). The province to the west of ISSG (EAFR) also has
a similar annual trend with similar concentrations to those observed in the ISSG province. The
slightly higher concentration in June in these two provinces is attributed to winter blooms
occurring off the Madagascar coast (Dilmahamod et al., 2019). This trend is different in the
AUSW province (east of ISSG), where the values are also higher during November (2.43 nM),

December (3.19 nM) and January (3.23 nM) but a clear low is observed during August (1.05 nM) (Figure S10).

In the southern hemispheric Atlantic Ocean sector, the provinces SATL, FKLD, BRAZ and
BENG are a part of the South Atlantic Gyre. The mean values in these provinces decrease from approximately 2 nM in January to less than 1 nM in May (Figure S10). The African coastal province, BENG shows higher values in January (4.09 nM) and March (3.59 nM) with lower concentrations in February (2.54 nM). This province does not show significant variation throughout the year owing to the continual supply of nutrient-rich water due to an upwelling
that supports primary production and hence DMS production (Jury and Brundrit, 1992) (Figure S10).

In the southern hemispheric Pacific Ocean sector, the provinces AUSE, TASM and NEWZ cover the eastern and southern coastal region of Australia and coastal New Zealand. The primary production in TASM and NEWZ shows an annual cycle controlled by the availability
of solar radiation, wind stress and MLD similar to most provinces but the variation is large for these variables   (Chiswell et al., 2013). Variability in the DMS concentrations follows variability in chlorophyll and sunlight with the highest concentrations observed in the summer, reducing in autumn with a decreasing trend until the summer of the following year. SSTC DMS concentrations are constant throughout the year, although from July to October DMS levels are
consistently lower. This is most likely due to the autumn, winter and spring phytoplankton blooms observed in the TASM and NEWZ regions (Chiswell et al., 2013) (Figure S10).

Provinces in the northern extra-tropical (BERS, PSAW, PSAE, NECS) typically show higher values in the northern hemispheric summer. The provinces in the North Atlantic extra-tropical gyre (NAST(W), NAST(E), GFST, NWCS, NECS and MEDI) typically show peaks during
the northern summer. This pattern results from a combination of plankton seasonal species

succession and short-term sunlight effects on DMS budgets, stimulating DMS production, inhibiting bacterial consumption, enhancing photolysis, etc. (Galí and Simõ, 2015). The provinces NECS and MEDI show a similar trend to the polar province SARC. In addition to the summer peaks, an additional peak is also observed during the northern spring that has

higher values than the peak in summer. The trends in the NECS province appear to be dominated by the spring blooms of the Baltic and the North Sea (April=5.56 nM; May=7.21 nM). A second bloom driven by dinoflagellates has been reported to occur in July (Hjerne et al., 2019) and coincides with a second peak in the DMS concentrations at 5.94 nM. The GFST and NWCS provinces show similar values and trends with twin maxima in spring and summer.

NWCS, being a coastal province, shows consistently higher values than the GFST province. These higher values in the North Atlantic provinces are mostly associated with the *Phaeocystis pouchetii* and coccolithophores blooms occurring in the months of May-July (Iida et al., 2002; Matrai et al., 2007; Matrai and Keller, 1993) (Figures S10 and 6).

In the Pacific region, the BERS, PSAW, and PSAE provinces show higher values during May

followed by lower concentrations in June and the concentrations rise again in July. A gradual decrease follows the summer peak, and the lowest values are observed during winter in December-January. In the Bering Sea and adjacent regions, blooms tend to form due to the dynamics of sea-ice retreat in the spring (Jin et al., 2007; Sigler et al., 2014). This most likely contributes to the maxima observed in the respective provinces. Although significant blooms

were also reported in the fall, (Ardyna et al., 2014) the magnitude of increase in the DMS concentrations is smaller after more data has been added.

### 3.1.4. Tropics

In the Indian Ocean sector of the tropics, the province MONS is in the central Indian Ocean with coastal provinces of REDS, INDW and INDE to the north, ARAB to the west, AUSW to

the east and ISSG to the south. The annual variation in the MONS province is like the ISSG

province, although with the peak value observed in February instead of December (3.61 nM).

The INDE and REDS provinces show a similar monthly variation as the ARAB province. This

region experiences the southwest and northeast monsoons during the periods between June to

September and October to December.  The Somali current flowing through the ARAB and the

INDE provinces is characterized by seasonal changes influenced by the southwest and

northeast monsoons. During June to September, the warm southwest monsoon moves the

coastal waters north-eastward, creating a coastal upwelling (Mccreary et al., 1996). This could

be a reason for the higher mean DMS concentrations observed in these provinces during June

and July (~2.8 nM). However, these variations in DMS are not observed in the INDW province

which is on the eastern coast of the Indian peninsula and not directly affected by the upwelling.

In November, the coastal Indian province INDW shows higher values than the ARAB province,

most likely due to the northeast monsoon winds increasing continental aerosol deposition over

the ocean surface, thereby increasing productivity due to increased nutrient availability

(Figures S10 and 6).

In the northern tropical Pacific Ocean, the province NPTG shows higher values in August (2.34

nM), while the lowest levels are observed in January (~1 nM). DMS concentrations increase

from a low in January to 1.61 nM in April, decreasing slightly to 1.29 nM in May and increasing

thereafter until August. In PNEC, there is no significant annual trend as the mean DMS

concentrations stay in the 2-2.6 nM range with one lower value in May (1.59 nM). From May

to August, PNEC and NTPG show an increase in DMS (Figures S10 and 6).

In the southern tropical Pacific region, the provinces SPSG, CHIL, PEQD and PNEC are seen

to have little variability unlike those seen in the Indian Ocean. The mean DMS values in the

SPSG province are higher in January and March (~3.6 nM), with a drop in February to 2.64

nM. The mean then drops to 1.38 nM in April followed by an increase to 2.74 nM in July. June

onwards, the mean DMS concentrations are ~2.6 nM (2.34 nM to 2.77 nM) up until December

(2.9 nM). During the southern summer months, CHIL is influenced by increased DMS

production with higher mean concentrations in December (4.91 nM) and January (7.53 nM).

This is also observed in the PEQD province as the mean DMS is slightly higher in December

(3.38 nM) and January (3.13 nM) compared to the other months. The Chile coastal 'Humboldt'

current is one of the most productive currents (Penven et al., 2005) and this may be driving the

higher DMS concentrations during May-June (May =4.27 nM, June=5.61 nM) in CHIL. Since

the current flows northward from CHIL to the province PEQD, the effect is seen as a slight

increase during June (3.35 nM) in PEQD (Figures S10 and 6).

The Atlantic Ocean tropical sector is comprised of the provinces CARB, NATR, CNRY,

ETRA, WTRA, GUIN, GUIA, and SATL. NATR and SATL are the major gyres in the northern

and the southern tropics. The gyres show elevated DMS concentrations during August (NATR)

and March (SATL). However, in the SATL province, the peak concentration is observed during

the transition from summer to autumn (March=2.64 nM). In the SATL province, the open ocean

sees the confluence of the Brazilian-Malvinas current which significantly affects the

productivity of the region. During most months, DMS concentrations are in the range of 1 –

2.5 nM in this province, with the lowest observed in May (0.83 nM). The NATR province has

higher values throughout the year compared to SATL, except in March. In the ETRA and

WTRA provinces, a clear annual peak is seen in August, which coincides with the maximum

extent and magnitude of phytoplankton blooms (July-September) in the eastern equatorial

Atlantic Ocean (Pérez et al., 2005). The inputs from the Amazon and Congo rivers are a major

source of nutrients in this area throughout the year, with peak input between May and

September. Similarly, in the GUIA and the CARB provinces, the Amazon and the Orinoco

river inputs influence the seasonal variability of primary production (Forget et al., 2011). This

is seen in the peak concentrations of DMS in CARB (September=2.74 nM) and GUIA

(August=2.75 nM). In the eastern Atlantic, the GUIN province shows a seasonal presence, with the CNRY province occupying its location as seen in the dynamic biogeochemical province map (Figure S1, S10). Three characteristic peaks are seen in the CNRY province in March (3.86 nM), August (3.34 nM) and November (2.45 nM) (Figures S10 and 6).

### 3.2. Differences due to usage of dynamic biogeochemical province boundaries

Dynamic biogeochemical province boundaries as defined by Reygondeau et al. (Reygondeau et al., 2013) were used in the DMS-Rev3 climatology instead of static boundaries to incorporate the monthly variability of surface properties in the world's oceans. In this section, we describe the differences caused by using the dynamic province boundaries as compared to the static boundaries (Figure 7 and S11a). As expected, most of the differences are seen at the province

boundaries when compared to the output using static provinces. Major differences are seen in the polar regions and smaller differences in the extra-tropical and tropical oceans. When globally averaged, the annual difference caused in the climatology by using the dynamic province boundaries is only about 0.02 nM (~5.1%) (Table S1), however, regional differences are up to ± 5 nM (±100%) (Figure 7 and S11a).

The sea-ice mask excludes data from the Arctic region when sea-ice cover exceeds 50%. If the sea-ice mask is ignored, the use of dynamic provinces generates significant differences in the Arctic Ocean region, with increases of 1-3 nM (+100%) during February-March and decreases of 1 to 3 nM (-40% to -60%) during April (Figure S11b). The winter months and spring months show a positive change in the DMS concentrations as compared to the summer months where

a reduction in the DMS concentrations is observed (Figure 7). The Arctic region, especially the Nordic Seas where ice-free regions exist even during the winter season, showed positive a difference with a maximum of 100% change during February. The difference stayed positive

during the northern winter (~40% to 60%) while it ranged from 0 to ~-40% for the northern summer.

The Southern Ocean region showed the greatest regional differences in November where concentrations differed by >4 nM (+100%), while the largest reduction is observed in January (-4 nM; -50%). Both large positive and negative differences are seen in this region during the summer months with a negligible difference during the rest of the year.

For the rest of the regions in the extra-tropics and tropics, regional differences of between ± 2
nM (up to 100%) can be seen, changing throughout the year. Most of the world's oceans show differences of ±20%. Clear patches with larger differences can be observed such as in the Atlantic Ocean where the differences are around -2 nM (-60% to -80%). A few patches are also observed in the northern and equatorial Pacific Ocean, where large differences are seen in different locations depending on the month (Figure 7). Globally, the largest difference is seen
during November, where the use of dynamical province boundaries leads to an increase of 0.36 nM in the globally averaged concentration, while the smallest difference is seen during May (-0.003 nM).

### 3.3. Differences due to usage of Variability Length Scale (VLS)

The other major change in DMS-Rev3 was the use of DMS VLS, which led to significant
changes, reducing the patchiness that was a feature of past climatologies. In this section, we present the effect of using VLS compared to the fixed value of 555 km when applying the Barnes filter (see Section 2.5). The differences ($DMS_{VLS}$-$DMS_{555}$) are shown in Figure 8 and the percentage differences (($DMS_{VLS}$-$DMS_{555}$) x 100/$DMS_{555}$) are shown in Figure S12. When globally averaged, the annual DMS concentration increased by 0.025 nM (+7.4%) when using
the VLS while applying the Barnes filter (Table S1). Annually, the largest differences of up to ±4 nM (±90%) were observed in the coastal Antarctic region and smaller increases were



observed in certain regions of the Atlantic and Pacific Oceans. However, over most of the world's oceans, the difference was within ±0.3 nM (±5%).

When the monthly differences are considered, large regional differences show up between the

VLS and 555 km filters. For instance, during the southern hemispheric summer (DJF), large differences of above ±5 nM (±100%) are seen in the Southern Ocean. This suggests that the use of a fixed value of 555 km led to severe over and underestimations in this region. During February, the entire Indian Ocean region shows patches of both positive and negative differences. The central Atlantic region also shows positive differences during the months from

March to May, with the largest differences observed in March ~2 nM (>100%). The Pacific Ocean shows the least differences amongst all the world's oceans. The Arctic region also shows differences of above 100%, especially during the summer season. Overall, the largest differences are observed in the high concentration polar regions, with smaller differences seen close to the tropics.

**3.4. Differences due to sea-ice cover**

Applying the sea-ice cover filter after the data unification before creating the DMS-Rev3 climatology, there is a marginal change in the global annual mean DMS (<1%) at the 50% threshold. The most noticeable changes are when the filter is applied after the creation of climatology. The global annual DMS reduces from 2.35 nM to 2.34 nM (0.4% reduction when

using a 95% sea-ice mask) and further down to 1.73 nM (~26% reduction when using a 20% sea-ice mask).

For this study, a threshold of 50% was considered for the sea-ice cover. The sea-ice cover removes a major part of the province of BPLR and parts of the BERS, SARC and ARCT were also masked reducing the monthly means of the respective provinces. Similarly, in the southern

region, the province APLR was also masked by the sea-ice. The southern hemisphere winter

shows the presence of sea-ice to a much larger extent covering areas of APLR and ANTA provinces.

### 3.5. Differences between DMS-Rev3 and L11

Figure 9 shows the differences between the DMS-Rev3 and L11 climatologies ($DMS_{REV3}$-$DMS_{L11}$) and Figure S14 shows the percentage difference (($DMS_{REV3}$-$DMS_{L11}$) x 100/$DMS_{L11}$) between the two. The most noticeable difference is the reduced patchiness in the data distribution. This is mainly due to the improvements made in the interpolation routine and the use of observed DMS VLS in the Barnes filter (Section 2.5). For the global annual climatology mean, 73% of the grid points show a positive difference, (Figure 9, S14) ~51% data shows differences up to ±1 nM (±10%), and ~81% of data has differences within ±2 nM (±30%) (Figure S14). Only 0.24% of data shows differences equal to or above 100% (>5 nM). Overall, the DMS-Rev3 climatology estimates a global annual average DMS concentration that is 0.05 nM less than the L11 climatology. The largest mean difference is seen in November (+0.31 nM) and the lowest difference is seen in September (-0.03 nM) (Table S1).

Most of the large differences are found in the polar regions, especially in the Southern Ocean. The equatorial oceans show differences less than ±0.5 nM, except in parts of the Indian and Pacific Oceans, where differences of up to -1 nM are observed (Figure 9). The two climatologies agree well in the oligotrophic regions of the oceans where lower DMS concentrations are observed. Monthly, the largest differences are seen in the Southern Ocean during the months of November-February, with the DMS-Rev3 climatology displaying higher values in November throughout the Southern Ocean. For the rest of the southern hemisphere summer, positive and negative differences of up to 5 nM are seen in the Southern Ocean, where the negative differences coincide with the patches that were a feature in the L11 climatology.

### 4. DMS Flux

The global sea-air DMS flux (Figure 10) reveals that ~93% of the world oceans emit DMS in

the range of 0-10 µmol S m$^{-2}$ d$^{-1}$ , with few hotspots in the world's coastal regions like BENG,

ARAB, etc. provinces along with oceanic provinces like the confluence of PEQD and SPSG in

the Pacific, ISSG province in central Indian Ocean region, which covers ~7% of oceans with

emissions in the range from  10 – 20 µmol S m$^{-2}$ d$^{-1}$. The Southern Pacific Ocean shows higher

values as compared to the Northern Pacific, although the Eastern Pacific shows more flux as

compared to the western Pacific mainly due to the coastal provinces of CHIL, CAMR, CCAL

and ALSK which show particularly large fluxes. The North Atlantic region, mainly in the

GFST and NADR, shows a higher flux compared to the South Atlantic. The equatorial Indian

Ocean region shows lower values than the rest of the Indian Ocean. The Arabian Sea shows a

higher flux of DMS compared to the Bay of Bengal. The total global annual DMS flux is

estimated at 27.1 TgS yr$^{-1}$, about 3.5% lower than the L11 climatology. DMS flux variability

follows the variability in DMS concentration, which is significant at the regional scale.

The entire oceanic region to the south of ~30° S seems to be a major source of DMS during the

southern hemisphere summer. The northern hemisphere summer shows elevated DMS

emissions in only a few regions. Additional regions are visible as hotspots in the open ocean

(e.g. in the North Atlantic and the Central Indian Ocean) as well as in coastal regions like those

observed off Chile coast, South of Alaskan coast in the Pacific Ocean and the North-western

Arabian Sea. The hotspot observed in the Central Indian Ocean region is present almost through

all seasons although it seems to be weakening during the northern spring. With an annual

average of ~2.76 nM, this region is above the global average, increasing the annual DMS flux

(~14 µmol S m$^{-2}$ d$^{-1}$).

Since the Nightingale et al parameterization (Nightingale et al., 2000) uses the wind speed and

the SST for estimation of the gas transfer from sea to air, the DMS flux hence calculated shows

a direct relationship with the wind speed and SST. However, wind speed is a major driving

force of the gas transfer velocity, which is a major determinant of the flux. As observed by Bell

et al and Zavarsky et al. (Bell et al., 2013; Zavarsky et al., 2018a), the wind-speed of about

$\sim>10$ m s$^{-1}$ seems to be one of the main factors to suppress the DMS fluxes. The optimum

response was observed in the range of 5-8 m s$^{-1}$ approximately comparing the wind speed

climatology and DMS fluxes in the region where the DMS fluxes were found to be higher over

the global oceans. This explains most of the regional flux variability as observed in the DMS-

Rev3 output. The evidence of this is seen in the Antarctic coastal region, which seems to be a

major source of DMS during the southern summer. But due to the higher wind speed over the

region, the contribution to the DMS flux is lower than one would expect. This region also has

lower SST which further reduces the gas transfer.

The differences in the fluxes with respect to L11 show a similar pattern to that seen in the

distribution of oceanic DMS on monthly and annual scales (Figure S15). This means that

although the change in the global annual average of DMS flux is not substantial, the DMS-

Rev3 climatology has large regional changes due to a reduction in the patchiness of the oceanic

DMS concentrations. Most of the high latitudes show a reduced flux, while the tropical regions

show an enhanced flux.



### 5. Conclusion

An updated global sea-surface DMS concentration climatology was created by upgrading the processing algorithm initiated by Kettle et al., (1999) and Lana et al., (2011), along with the inclusion of new data compiled from various sources. The global annual average concentration reduced to 1.87 nM, although large differences of up to 5 nM were observed on regional scales during certain months. This is an important difference considering the effect of regional emissions on the total impact of DMS on the Earth's radiative budget (Fiddes et al., 2018; Mahajan et al., 2015; Thornhill et al., 2020; Woodhouse et al., 2013). The global sea-air flux of DMS is estimated at 27.1 TgS yr$^{-1}$ which is similar to L11 (a 3.5% decrease). The use of dynamic province boundaries allowed the estimation of more realistic annual trends in different regions. The patchiness in the climatology identified in previous estimates was reduced by region-specific data exclusion (Section 2.2) and the usage of the observation-based VLS (Section 2.5). The use of dynamic province boundaries and VLS was not important at the annual and global scales but resulted in large differences at regional scales. Although this climatology shows significant improvements in the estimation of seawater DMS concentrations, it still suffers from a lack of continuous observations, especially in certain parts of the world's oceans. Focus can be given specifically to the provinces NEWS, AUSE, AUSW, SUND and ARCH which are present around Australia including the Great Barrier Reef, and the Indonesian archipelago. Along with these regions, the northern and eastern regions of the Arabian sea (EAFR, INDE and REDS provinces) also need extensive sampling for reducing the dependence on substitution and interpolation. Another major uncertainty is the contribution in regions affected by sea-ice. DMS-Rev3 essentially removes data from sea-ice regions according to a fixed percentage, neglecting the contribution of these regions. Thus although this new climatology is a major upgrade from the past estimates, and matches the estimates





using top-down methods, further improvements are needed in the future, with the main limiting

factor being data availability.

**Acknowledgements and Data**

IITM is funded by the Ministry of Earth Sciences (MOES), Government of India. T.G.B.

contribution to this study was supported by the NASA North Atlantic Aerosols and Marine

Ecosystems Study (grant no. NNX15AF31G). G.M. (with input from T.G.B. and P.H.)

contributed the VLS analysis, which is part of his PhD (NERC industrial CASE studentship

NE/R007586/1).

The data used for creating the climatology, along with the algorithm can be found in the online

repository: (Mahajan, 2021), "Third Revision of the Global Surface Seawater Dimethyl Sulfide

Climatology (DMS-Rev3)", Mendeley Data, V1, doi: 10.17632/hyn62spny2.1



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

**Tables and figures**

**Table 1**: Details of the number of raw data points, the threshold used for filtering extreme values, number of data points after binning, number of months with data in each biogeochemical province, before and after substitution are given. The provinces used as a donor where substitution was made are also listed. Provinces that were substituted (with data in less

than five months) are highlighted in bold.

| Province number | Code | Province name | Biome | Ocean | Threshold according to 99.9 Percentile (nM) | No. of raw data points | | No. of data lost due to filtering | Binned data before substitution | | Substituted by which province | Binned data after substitution | |
|---|---|---|---|---|---|---|---|---|---|---|---|---|---|
| | | | | | | Before filtering | After filtering | | No. of data | No. of months | | No. of data | No. of months |
| 1 | ETRA | Eastern tropical Atlantic | Trades | Atlantic | 8.8 | 126 | 125 | 1 | 234 | 6 | | 234 | 6 |
| 2 | APLR | Austral polar | Polar | Antarctic | 344.1 | 327565 | 327237 | 328 | 9790 | 8 | | 9790 | 8 |
| 3 | ANTA | Antarctic | Polar | Antarctic | 80.3 | 325400 | 325075 | 325 | 7816 | 9 | | 7816 | 9 |
| 4 | SANT | Subantarctic water ring | Westerly | Antarctic | 8.0 | 45205 | 45160 | 45 | 3762 | 9 | | 3762 | 9 |
| 5 | SSTC | South subtropical convergence | Westerly | Antarctic | 19.9 | 17543 | 17525 | 18 | 2420 | 11 | | 2420 | 11 |
| 6 | WTRA | Western tropical Atlantic | Trades | Atlantic | 7.0 | 382 | 381 | 1 | 654 | 8 | | 654 | 8 |
| 7 | NEWZ | New Zealand coast | Coastal | Pacific | 28.5 | 1465 | 1464 | 1 | 930 | 5 | | 930 | 5 |
| **8** | **AUSE** | **East Australian coast** | **Coastal** | **Pacific** | **8.6** | **36** | **35** | **1** | **44** | **3** | **AUSE + NEWZ** | **758** | **7** |
| **9** | **SUND** | **Sunda-Arafura shelves** | **Coastal** | **Pacific** | **6.9** | **777** | **775** | **2** | **416** | **4** | **SUND + AUSE** | **426** | **5** |
| 10 | NATR | North Atlantic tropical gyre | Trades | Atlantic | 14.6 | 1749 | 1747 | 2 | 2526 | 12 | | 2526 | 12 |
| 11 | CHIN | China Sea | Coastal | Pacific | 16.4 | 830 | 829 | 1 | 700 | 8 | | 700 | 8 |
| 12 | CHIL | Humboldt current Chile coast | Coastal | Pacific | 80.3 | 796 | 795 | 1 | 958 | 6 | | 958 | 6 |
| **13** | **CAMR** | **Central American coast** | **Coastal** | **Pacific** | **5.4** | **130** | **129** | **1** | **112** | **1** | **CAMR + CCAL** | **4966** | **8** |
| 14 | CCAL | Coastal Californian current | Coastal | Pacific | 42.1 | 10981 | 10970 | 11 | 4854 | 7 | | 4854 | 7 |
| 15 | ALSK | Alaska coastal downwelling | Coastal | Pacific | 154.9 | 4659 | 4654 | 5 | 4824 | 9 | | 4824 | 9 |
| **16** | **ARCH** | **Archipelagic deep basins** | **Trades** | **Pacific** | **7.3** | **166** | **165** | **1** | **50** | **3** | **ARCH + TASM** | **326** | **6** |
| 17 | WARM | Western Pacific warm pool | Trades | Pacific | 8.5 | 10429 | 10419 | 10 | 2174 | 6 | | 2174 | 6 |
| 18 | PEQD | Pacific equatorial divergence | Trades | Pacific | 44.9 | 1395 | 1394 | 1 | 1682 | 10 | | 1682 | 10 |
| 19 | PNEC | North Pacific equatorial | Trades | Pacific | 6.9 | 11325 | 11314 | 11 | 1766 | 10 | | 1766 | 10 |

| | | | | | | | | | | | | | |
|---|---|---|---|---|---|---|---|---|---|---|---|---|---|
| | | counter current | | | | | | | | | | | |
| 20 | NPTG | North Pacific Tropical gyre | Trades | Pacific | 6.6 | 8129 | 8121 | 8 | 1270 | 11 | | 1270 | 11 |
| 21 | NASTW | Northwest Atlantic subtropical gyre | Westerly | Atlantic | 8.6 | 1281 | 1280 | 1 | 1616 | 10 | | 1616 | 10 |
| 22 | SPSG | South Pacific gyre | Trades | Pacific | 11.4 | 10539 | 10528 | 11 | 4768 | 8 | | 4768 | 8 |
| **23** | **TASM** | **Tasman Sea** | **Westerly** | **Pacific** | **6.8** | **3996** | **3992** | **4** | **422** | **4** | **TASM + ARCH** | **448** | **6** |
| 24 | C(O)CAL | California current | Trades | Pacific | 48.4 | 4811 | 4806 | 5 | 2850 | 8 | | 2850 | 8 |
| **25** | **NPSW** | **Northwest Pacific subtropical** | **Westerly** | **Pacific** | **5.0** | **146** | **145** | **1** | **196** | **4** | **NPSW + NPSE** | **264** | **7** |
| 26 | NPSE | Northeast Pacific subtropical | Westerly | Pacific | 8.3 | 130 | 129 | 1 | 192 | 6 | | 192 | 6 |
| 27 | NPPF | North Pacific polar front | Westerly | Pacific | 16.9 | 251 | 250 | 1 | 378 | 7 | | 378 | 7 |
| **28** | **KURO** | **Kuroshio current** | **Westerly** | **Pacific** | **10.8** | **697** | **696** | **1** | **728** | **4** | **KURO + NPSE** | **788** | **6** |
| 29 | PSAW | Western Pacific subarctic gyre | Westerly | Pacific | 11.0 | 252 | 251 | 1 | 290 | 5 | | 290 | 5 |
| 30 | PSAE | Eastern Pacific subarctic gyre | Westerly | Pacific | 27.9 | 3043 | 3040 | 3 | 1288 | 5 | | 1288 | 5 |
| 31 | BERS | North Pacific epicontinental sea | Polar | Pacific | 151.0 | 3856 | 3852 | 4 | 3928 | 9 | | 3928 | 9 |
| 32 | GFST | Gulf Stream | Westerly | Atlantic | 16.1 | 2073 | 2071 | 2 | 1460 | 11 | | 1460 | 11 |
| 33 | NADR | North Atlantic Drift | Westerly | Atlantic | 27.1 | 4706 | 4701 | 5 | 3224 | 10 | | 3224 | 10 |
| **34** | **AUSW** | **Western Australian and Indonesian coast** | **Coastal** | **Indian** | **3.1** | **5997** | **5991** | **6** | **314** | **3** | **AUSW + AUSE + SUND** | **488** | **6** |
| 35 | INDE | Western India coast | Coastal | Indian | 220.5 | 98 | 97 | 1 | 176 | 8 | | 176 | 8 |
| **36** | **INDW** | **Eastern India coast** | **Coastal** | **Indian** | **6.8** | **98** | **97** | **1** | **186** | **3** | **INDW + ARAB** | **322** | **8** |
| 37 | ARAB | Northwest Arabian Sea upwelling | Westerly | Indian | 16.9 | 150 | 149 | 1 | 268 | 8 | | 268 | 8 |
| **38** | **REDS** | **Red Sea, Arabian Gulf** | **Coastal** | **Indian** | **2.0** | **14** | **13** | **1** | **26** | **1** | **REDS + ARAB** | **266** | **8** |
| **39** | **EAFR** | **Eastern India coast** | **Coastal** | **Indian** | **20.1** | **303** | **302** | **1** | **76** | **2** | **EAFR + ISSG** | **1112** | **7** |
| 40 | ISSG | Indian South subtropical gyre | Trades | Indian | 12.5 | 13574 | 13560 | 14 | 1738 | 7 | | 1738 | 7 |
| 41 | MONS | Indian monsoon gyre | Trades | Indian | 12.1 | 352 | 351 | 1 | 586 | 7 | | 586 | 7 |
| 42 | SARC | Atlantic sub-Arctic | Polar | Atlantic | 66.7 | 1296 | 1295 | 1 | 1702 | 6 | | 1702 | 6 |
| 43 | BENG | Benguela current coast | Coastal | Atlantic | 13.4 | 6399 | 6393 | 6 | 652 | 5 | | 652 | 5 |
| **44** | **FKLD** | **Southwest Atlantic shelves** | **Coastal** | **Atlantic** | **18.9** | **168** | **167** | **1** | **276** | **4** | **FKLD + BRAZ** | **364** | **7** |
| 45 | BRAZ | Brazilian current coast | Coastal | Atlantic | 23.7 | 1060 | 1059 | 1 | 296 | 7 | | 296 | 7 |
| 46 | ARCT | Atlantic Arctic | Polar | Atlantic | 19.0 | 6311 | 6305 | 6 | 3572 | 9 | | 3572 | 9 |



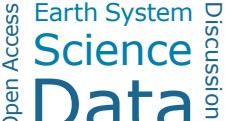

| 47 | NASTE | Northeast Atlantic subtropical gyre | Westerly | Atlantic | 19.7 | 5889 | 5883 | 6 | 4126 | 12 | | 4126 | 12 |
| 48 | CARB | Caribbean | Trades | Atlantic | 23.0 | 992 | 991 | 1 | 1318 | 9 | | 1318 | 9 |
| 49 | MEDI | Mediterranean Sea | Westerly | Atlantic | 60.5 | 625 | 624 | 1 | 864 | 11 | | 864 | 11 |
| 50 | NWCS | Northwest Atlantic shelves | Coastal | Atlantic | 28.6 | 3329 | 3326 | 3 | 3102 | 11 | | 3102 | 11 |
| **51** | **GUIA** | Guianas coast | **Coastal** | **Atlantic** | **2.3** | **8** | **7** | **1** | **12** | **1** | **GUIA + WTRA** | **432** | **8** |
| **52** | **GUIN** | Guinea current coast | **Coastal** | **Atlantic** | **NaN** | **0** | **0** | | **0** | **0** | **ETRA** | **234** | **6** |
| 53 | CNRY | Canary current coast | Coastal | Atlantic | 32.0 | 73 | 72 | 1 | 138 | 6 | | 138 | 6 |
| 54 | NECS | Northeast Atlantic shelves | Coastal | Atlantic | 198.5 | 2661 | 2658 | 3 | 4376 | 11 | | 4376 | 11 |
| 55 | SATL | South Atlantic gyre | Trades | Atlantic | 6.7 | 14072 | 14058 | 14 | 1866 | 9 | | 1866 | 9 |
| 56 | BPRL | Boreal polar | Polar | Arctic | 19.3 | 4006 | 4002 | 4 | 2970 | 8 | | 2970 | 8 |

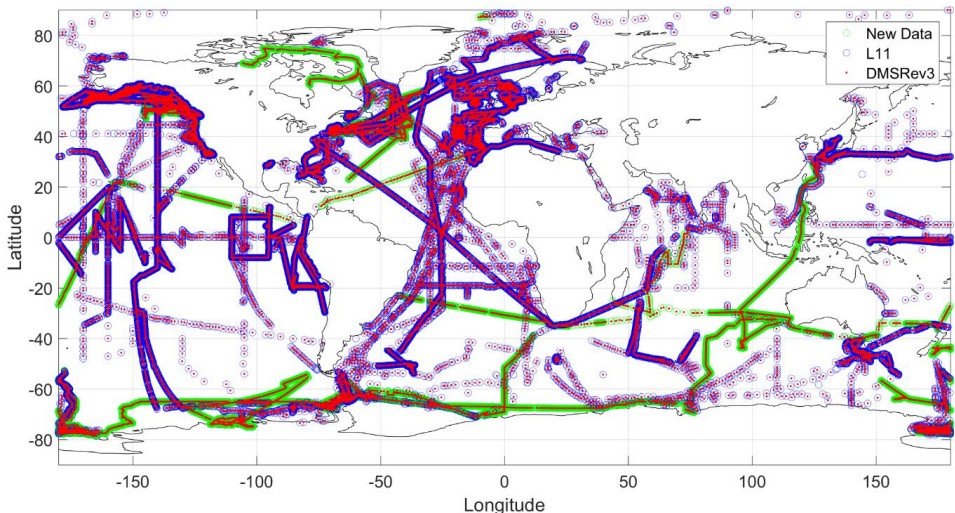

957

**Figure 1:** Data from different sources was put together for creating the raw input dataset, which

consisted of 872, 427 points. The data in green represent the new data, and the blue circles

represent data that was used in the L11 DMS climatology. Post quality control, and data

unification for addressing temporal and spatial sampling biases, 48,567 data points were used

as an input for the DMS-Rev3 climatology calculation algorithm (red dots).

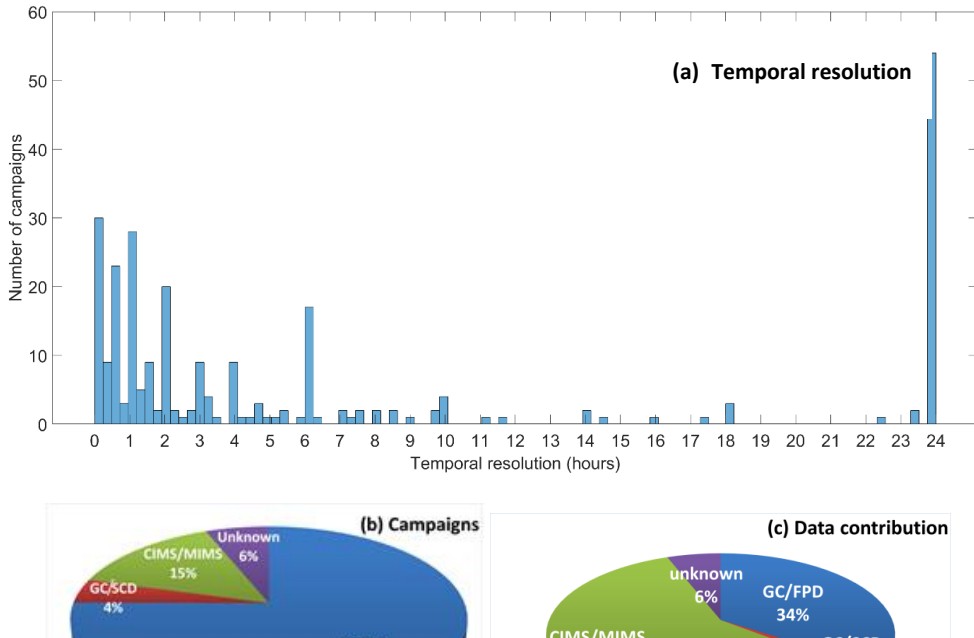

**Figure 2:** (a) Frequency distribution of the sampling interval for the campaigns included in this study are shown. (b) The type of measurement technique used to measure DMS during the individual campaigns. The 'unknown' dataset/campaigns resemble the frequency of a GC instrument. (c) The number of raw data points according to the measurement technique is shown. The number of data is dominated by CIMS/MIMS based measurements, but the number of campaigns is dominated by the GC measurements.

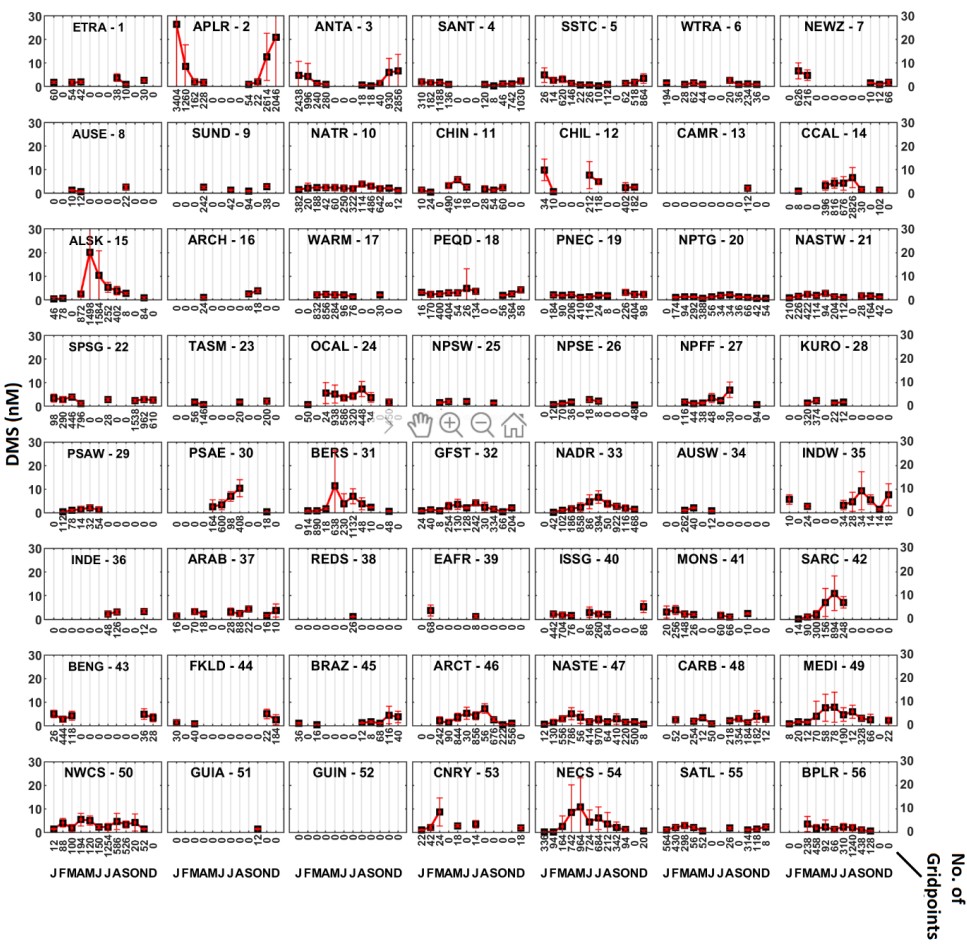

**Figure 3:** The data distribution in different biogeochemical provinces is different owing to the time, location, and frequency of observation. The black squares represent the monthly mean of data with standard deviations shown as error bars. The number of hourly binned observations in a month is shown below the *x*-axis.

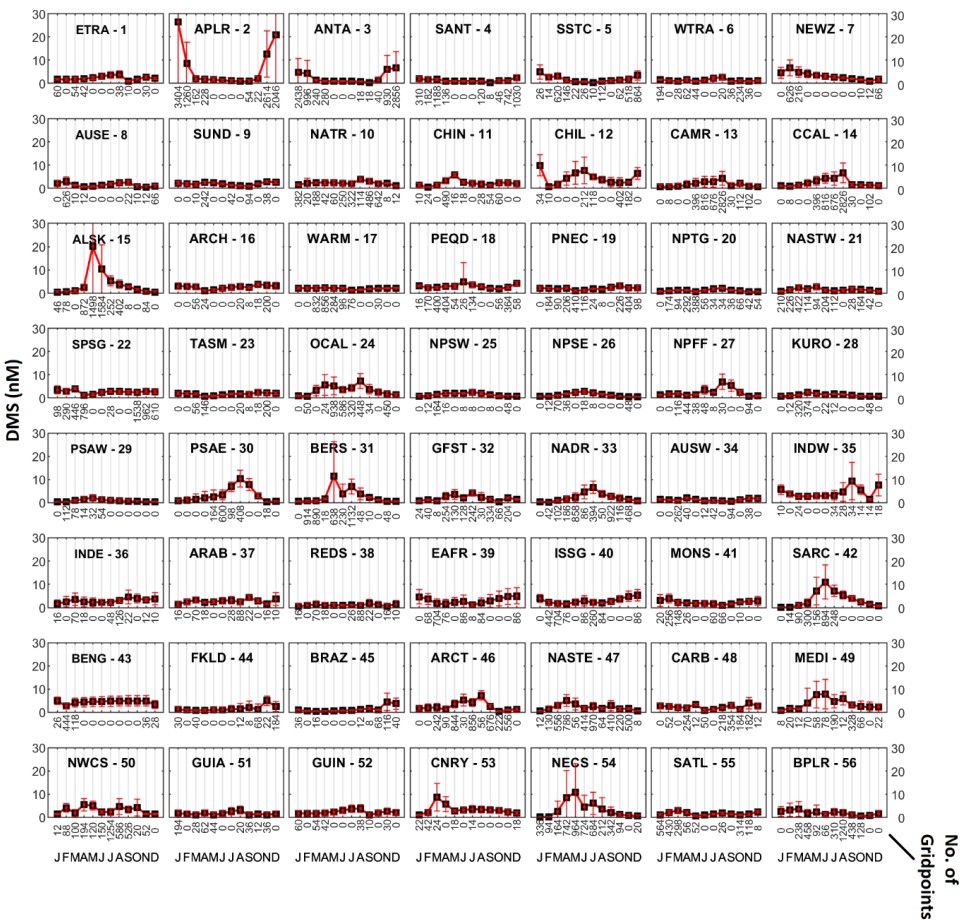

**Figure 4:** The substituted and interpolated monthly mean data (black squares) with respective

standard deviations for each province are shown. The number of hourly binned observations in

a month is shown below the *x*-axis; where 0 indicates that the data was interpolated.

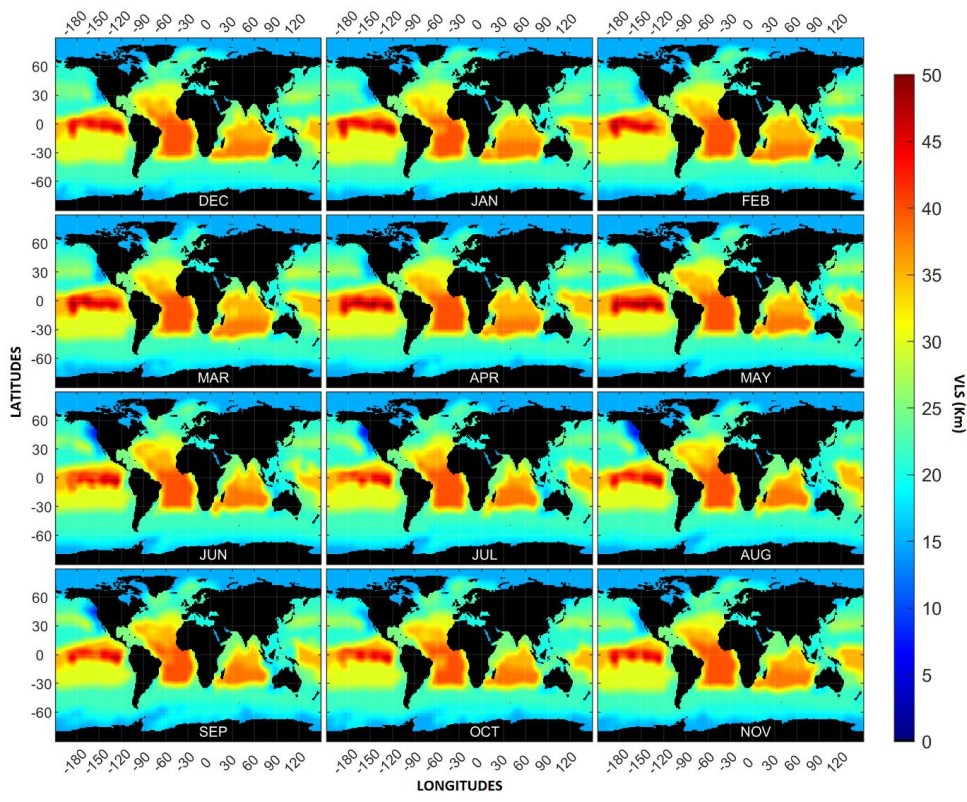

**Figure 5:** The DMS Variability Length Scale (VLS) used for the weighted-average interpolation computation is shown. The VLS in 11 provinces were based on past observations, and the VLS in the other provinces was estimated by looking for biogeochemical similarities.

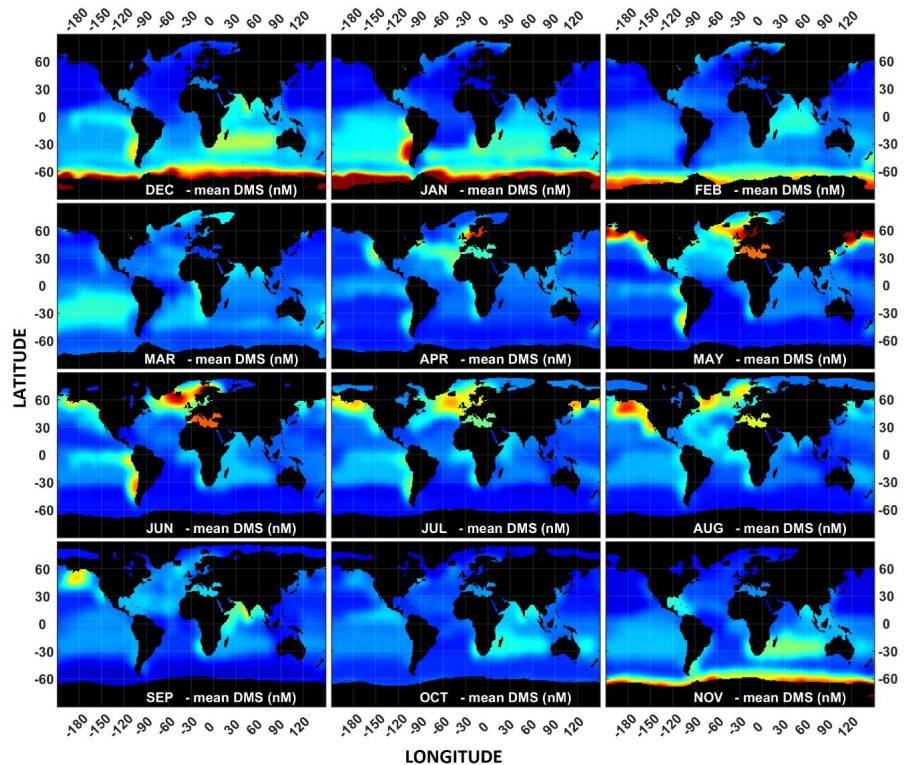

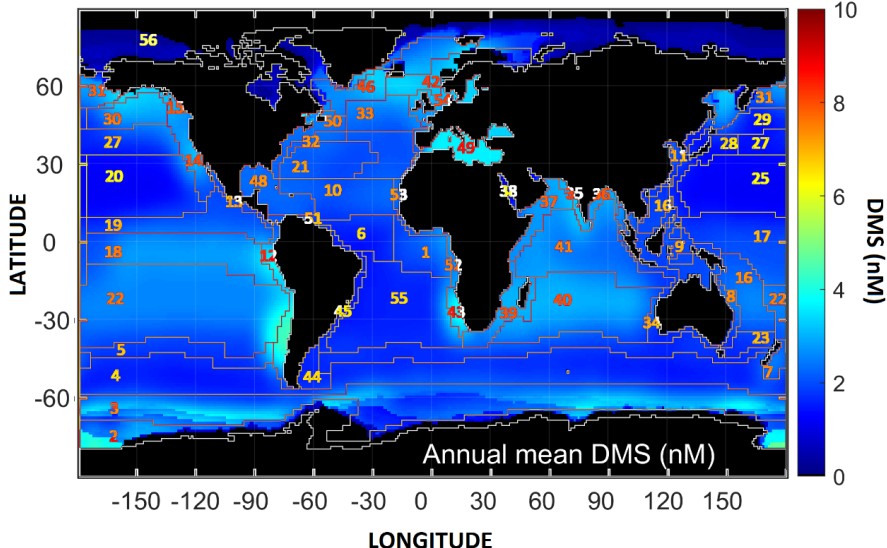

**Figure 6:** Distribution of the monthly and annual DMS concentrations as estimated by DMS-

Rev3 climatology with a 50% sea-ice mask.

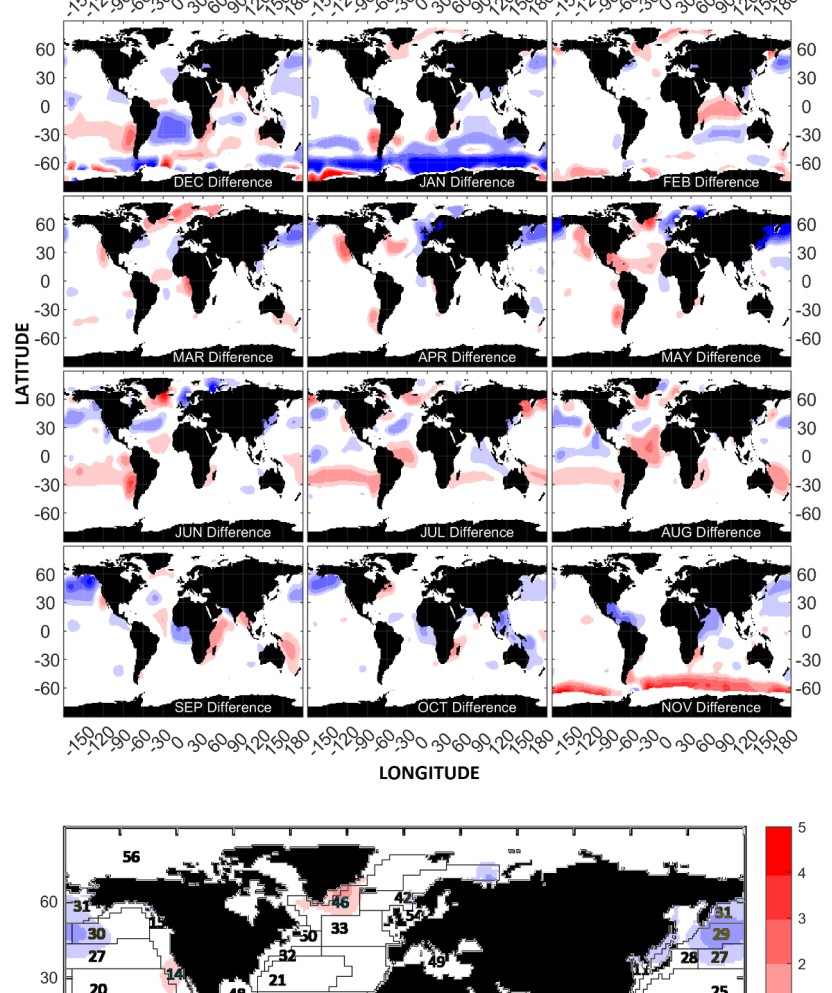

**Figure 7:** Differences between the monthly and annual mean DMS concentrations estimated

using dynamic and static biogeochemical province boundaries (Dynamic - Static).

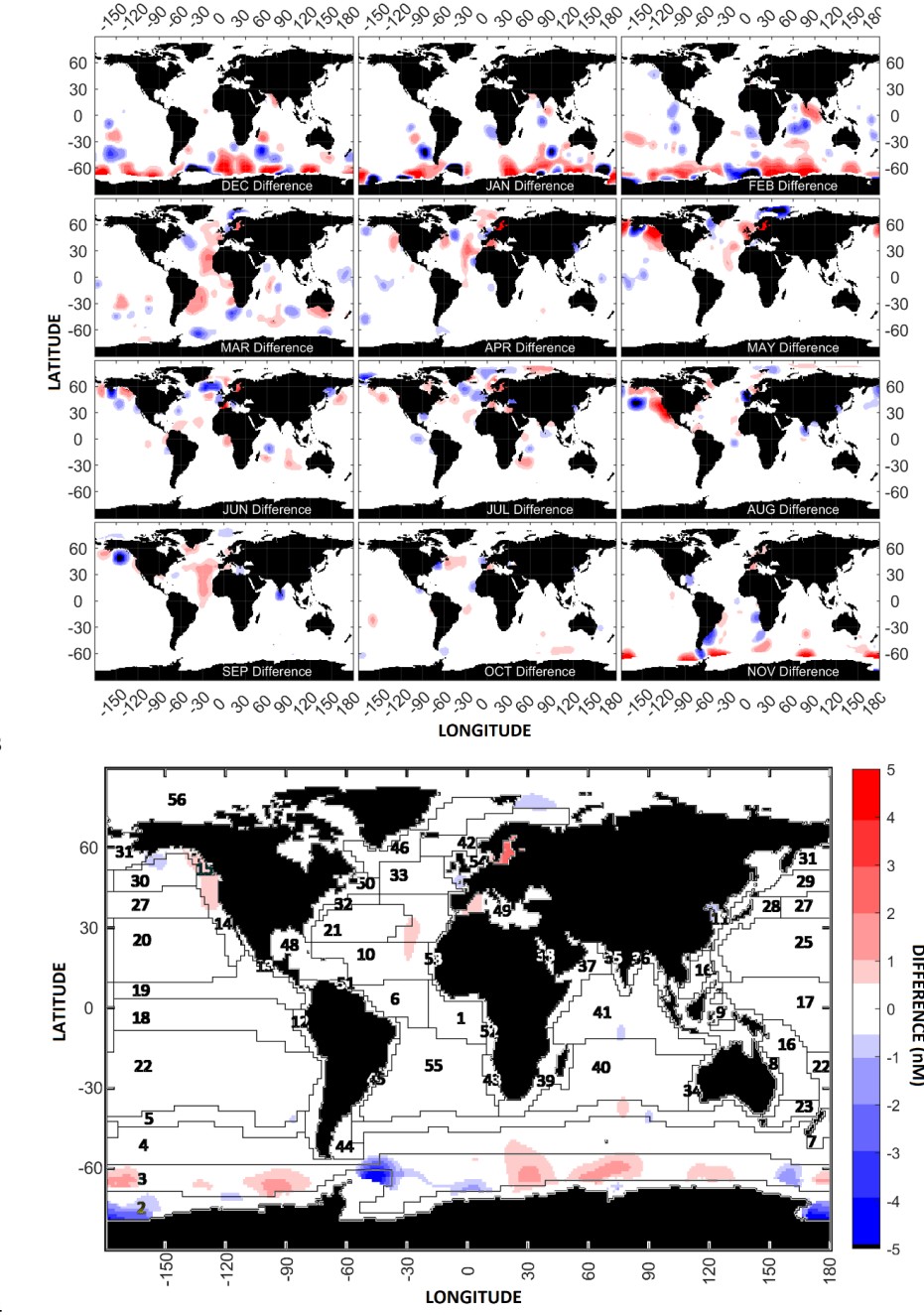

**Figure 8:** Differences in the DMS concentrations caused by the use of Variability Length Scale

(VLS) instead of a fixed value for the Radius of Influence as used by L11 (555 km) for the weighted

average interpolation computation. (VLS- 555 km).

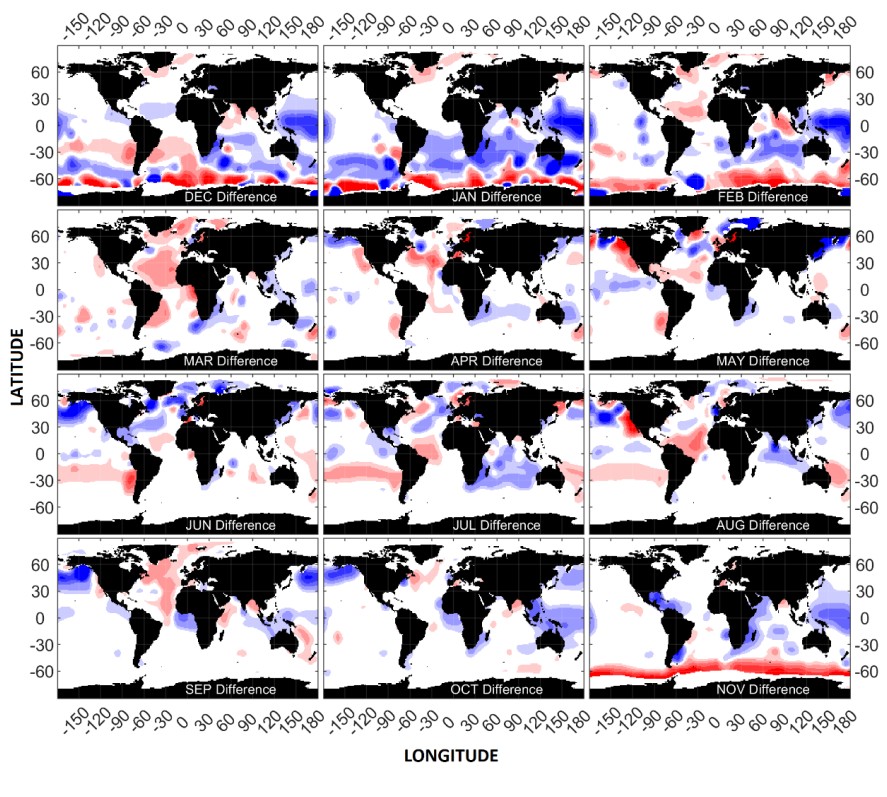

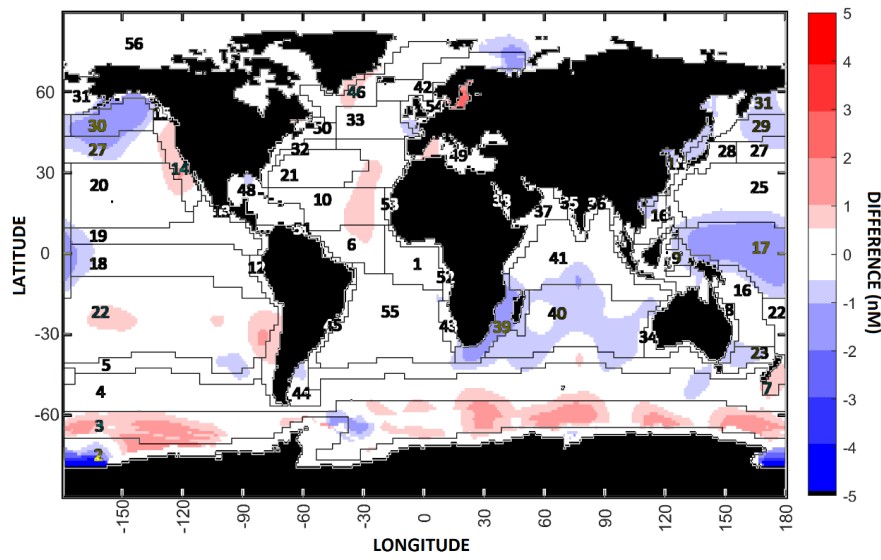

**Figure 9:** Differences between the monthly and annual mean DMS concentrations of the Rev3

and L11 climatologies (Rev3-L11).

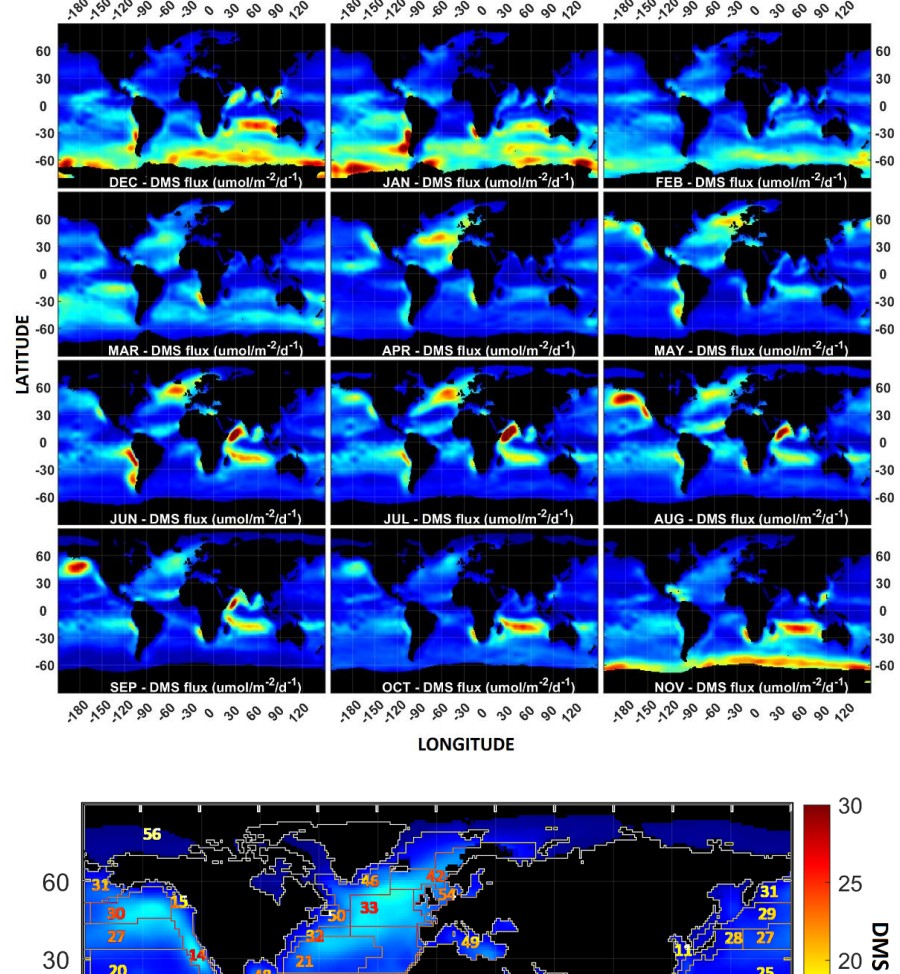

**Figure 10**: Distribution of the monthly and annual DMS flux as estimated by DMS-Rev3

climatology with 50% sea-ice mask.