# Peer review of "Third Revision of the Global Surface Seawater Dimethyl Sulfide Climatology (DMS-Rev3)"

_Earth System Science Data, 2021_

## Referee Comment (RC2)

[referee-annotated manuscript omitted]

---

## Author Comment (AC1)

**Response to reviewers' comments for manuscript number: essd-2021-236**

Comments by reviewers are shown in an italic typeface and the responses are shown in a normal typeface. Corresponding changes have been made in the revised manuscript and are marked in red.
* * *
*RC1: 'Comment on essd-2021-236', Giuseppe M.R. Manzella, 21 Oct 2021*

*Comments to:*

*Third Revision of the Global Surface Seawater Dimethyl Sulfide Climatology (DMSRev3)*

*by Hulswar et al.*

*The paper is well written and easy to follow. The description of mechanisms producing DMS is very clear also for non-expert people. The new climatology includes new data, and this is a paper of added value. Weak points are related to statistics, number of useful data for climatology, spatial and temporal distribution. These problems that are presented by the authors but not resolved.*

**Response**: We thank Dr. Manzella for the above comments and the detailed review. We have made changes to the manuscript to address the weaknesses as suggested. We hope that the modified manuscript will resolve any outstanding issues.

*The initial data set consisted of 872,427 data points of which only 48,567 are used after post processing. Therefore, the spatial and temporal coverage is worse than that shown in figure 1. Hence a first series of questions:*

- *In each month how many data are available in all geographical areas of 1 ° x 1 °? Does each square of 1x1 have a statistically significant number of data points? Are the data for each month and 1x1 areas statistically sufficient or should authors examine them seasonally?*

**Response**: As explained in the section 2.2 'Data Unification', the data was binned hourly which helped remove the sampling frequency bias while reducing the number of points. 48,567 points were not selected from the total 872,427 points but are a result of the hourly averaging of 872,427 points (873,539 after including a couple of new cruises since the paper was first submitted, which resulted in 48,898 datapoints after data unification) (line number 215-220). The number of datapoints in each province after this process are given in the figure 3 and figure 4 on the x-axis. In figure 3, the label '0' on the x-axis indicates absence of data. As the figure

shows, several of provinces lack data during certain months, which is why there was a need to interpolate at the province level to create the first guess fields (result in Figure 4).

Regarding each 1x1 degree grid box, there is a lack of continuous measurements in every grid box, which is why we had to generate first guess fields on a province level. Later, information from grip points at the 1 x 1 degree helps estimate variations within each province. One would not expect data in each grid point, which is why there was a need to interpolate along both geographical and temporal scales.

- *The climatology obtained in data-poor areas with similarity-estimated VLS is not convincing, by taking into account that the phenomena under investigation are occurring at high frequency and varying from place to place.*

**Response**: The VLS distribution that we used was calculated in 11 of provinces where high-frequency measurements were available. The VLS in other provinces was estimated based on biogeochemical similarities with these 11 provinces. This was the logical 'next step' to improve the climatology, and while it might not be exact, it is the best estimate for the variance in DMS we have from observations and is a significant improvement compared to assumptions made in the last climatology. We have added a sentence in the manuscript clarifying this (line number 333-337).

- *Another point that the authors present but do not investigate is related to methodologies and technologies for data collection. Over the years they have changed and so has the data accuracy. In the paper there should be an indication of what is the final accuracy of the climatology in the various areas.*

**Response:** This is a valid point but unfortunately no robust criteria or accepted method for the selection or elimination of historical data. Intercomparison studies show that the data is expected to be within a range of ±25% of the reported value. We have mentioned in section 2.1 (lines 175-200) that the only filters possible to apply for data across decades are for the extreme values, for which we have improved the filtration as compared to the last climatology.

- *The authors should also provide information on calibration standard if exists and if used in data selection.*

**Response:** Discussing the calibration standards for each individual study from which data is collected is beyond the scope of this paper. We do not perform any calibration related selection for the data.

*The problem of ecological provinces is well posed, and the authors refer to previously published articles. The 'geographically homogeneous' data can be identified with a cluster analysis. In a non-static environment, it is possible that geographic homogeneity may vary over time. Authors should discuss this.*

**Response:** We agree that the geographical homogeneity will vary with time, which is why we chose to follow the work of Reygondeau et al., 2013 using the dynamic-changing biogeochemical provinces since it incorporates the variability over space and time to create (line numbers 240-250).

*The new data included in the paper makes it interesting and publishable after major revision.*

**Response:** We thank the reviewer for the above comment and hope that the changes we have made are now acceptable.

*Special comments.*

*Figure 1 should shows the total raw data (1a) and those used for climatology (1b).*

**Response:** Figure 1 shows both total raw data and the one used for climatology. The blue and the green circles are the raw data that was used, and the red dots superimposed on them is the data used for climatology. This shows that the global coverage did not change due to the data unification methods followed in the paper. We have also added a figure in the supplementary text showing only the raw data as requested (Figure S1).

*An indication of errors or accuracy in the various regions would be desirable*

**Response:**

The standard deviation for the monthly and annual climatology across the different geographical regions is provided in the supplementary material Figure S7.
* * *
*RC2: 'Comment on essd-2021-236', Patricia Matrai, 08 Nov 2021*

*This manuscript presents the 3rd version of the climatology of marine DMS concentration and flux estimates.*

*Having sampled DMS since the 1980s and provided data to the NOAA PML database over the past 20+ years, it is both rewarding and sobering to see what the spatial and temporal data distribution is; still scarce and often nonexistent for a compound that affects ocean biogeochemistry and ocean-air-clouds-climate processes. Even if DMS's piece of the climate action continues to decrease w/r to the initial CLAW \*hypothesis\*, DMS continues to provide the main path for CCN growth.*

*The ms is well written, clear and to the point. The main change is in polar regions; while the sea ice mask results in a better representation of DMS concentrations in the Southern Ocean, the 50% sea ice mask "eliminates" most of the Arctic- sub-Arctic transition from the climatology. In satellite oceanography, one uses a 15% sea ice cut off; even then, one loses all the coastal regions. It would be good to have a bit more discussion of what happens at varying levels of sea ice mask (rather than none, 100%, and the 50% middle).*

*Please note that I included comments pertinent to the SI file in the main ms file to clarify several figure legends.*

**Response**: We thank Dr. Patricia Matrai for the above comments. We observed that the annual global mean DMS concentration obtained from DMS-Rev3 shows a linear response to increasing or decreasing percent sea-ice. We have now included a figure showing the response of the averaged global DMS concentration to the % sea-ice filter used in the supplementary text (Figure S17) and discussed this further in the manuscript (Line 634).

The changes related to the Figure legends have now been made (Figure S10 and S11)
* * *
*RC3: 'Comment on essd-2021-236', Murat Aydin, 12 Nov 2021*

*This is the third revision of global surface seawater DMS climatology. DMS is an important natural source of atmospheric sulfur. The new climatology includes a substantial amount of new data and represents a significant upgrade over the previous version; it is likely to be widely used in various modeling applications. Overall, it is a reasonably well-written manuscript with sound methodology, although some aspects of the data processing were difficult to understand. My concerns, questions, and suggestions are explained in detail below, followed by more specific comments, most of which are fairly trivial.*

*One overarching concern I have is with regards to the decisions made during the initial stages of the data processing. I understand that subjective decisions are unavoidable when it comes to data filtering, but proper justification is important and the impact of these decisions should be quantified if possible.*

**Response**: We thank Dr. Murat Aydin for the comments and the detailed review. We have made changes to the manuscript to address the weaknesses as suggested. We hope that the modified manuscript will resolve any outstanding issues.

*First data "clean-up" decision is the omission of the near zero and negative measurements. I find it difficult to believe that the negative values are incorrectly reported numbers (Ln 182). Negative measurements, unless they are extreme negative numbers, commonly occur when working at or below detection limits due to uncertainties in the calibration and/or a system blank that is being subtracted from the signal. I would expect there to be roughly an equal number of small positive measurements that balance out the negatives so that the average is equal to zero within the uncertainties. I do not see a reason to throw out these data as doing so will introduce a positive bias to the data set, especially if data filling schemes are implemented after data removal. In the current case, I realize that a very small fraction of the data is thrown out and the impact of this data clean-up is likely insignificantly small. Still, I do not find the justification convincing. Do the negative numbers cause problems with the data processing that follows? A more elegant solution might be to replace all data below detection limit with zeros, or a very small positive number if zeros cause problems too.*

**Response**: While we agree that filtering the negative values can lead to a positive bias (although the reviewer is right is identifying that this is a minor fraction of the total data), negative DMS concentrations are not possible. To test the effect of removing this data, we did an additional computation of the DMS climatology by fixing the negative numbers to a value of 0.001 nM, (the lower detection limit for some high-end instruments) instead of deleting the negative numbers. This resulted a minor change in the resultant DMS climatology (difference was less than 0.001%) across different geographical locations where the negative measurements occurred. Hence both methods can be considered. We have clarified this in the manuscript (line 184).

*The second item is filtering out high numbers. I was glad to see they did not go with the L11 method of using a 99.9% filter on the entire data set. I think the biggest issue with that approach is it will filter out much more than 0.1% of the data from provinces with high DMS*

*concentrations. The question I have is, is there really a need to filter out any data at all? If you think that the highest 0.1% from each province do not represent surface ocean concentrations, it is necessary to filter out these measurements, I agree. However, if you think these high end measurements might represent blooms (Ln 186), I don't think it is justified to throw these data out, even if there are repeat samplings of the same bloom. My guess is blooms might be severely underrepresented in these data sets because I suspect they occupy a small fraction of the surface ocean, or is there evidence to the contrary?*

**Response**: The 99.9-percentile cut-off takes care of the extreme values that are present in the dataset, which could result due to many issues, including possible sampling in the bloom region. These extreme values could add bias in the monthly first guess means of the related provinces due to the low number of data in each pixel. As the reviewer has suggested, the blooms are underrepresented in the datasets, and hence including only a few high values would give geographical abnormalities in areas where observations were available only during blooms. Hence, the subjective best-choice was to filter to outliers. We have mentioned the shortcoming of this method as identified by the reviewer in the manuscript (Line 197).

*The next critical data handling decision is how to combine the newer high resolution mass spectrometer (HRMS) data sets with the older low temporal resolution GC data. Rather, the 1 hr averaging applied to the HRMS data before data unification. This decision should be primarily data driven and the related discussions should include two important criteria: 1) The fidelity of the HRMS data to real variability in the surface ocean. How much temporal averaging to HRMS data is needed to filter out the noise from issues related to these instruments and sampling methods to capture the variability in the surface ocean? 2) The relative representativeness of HRMS vs. GC data. Are the GC data representative of a larger volume of seawater averaged over a longer period than the HRMS? The paper does not discuss these points. Instead, there is an emphasis on the need to balance the influence of historical vs. modern data on the climatology. What is the reason to worry about this balance? This is not adequately explained. It is possible that the quantitative impact of this 1 hr averaging is not all that large, but it would be good to know because HRMS data will become even more dominant in the future.*

**Response**: We temporally averaged the data into hourly bins, the reason primarily being removal of bias towards the HRMS datasets, which would bias the climatology purely based on the total number of data available. The issue is not the amount of seawater sampled, but rather how much the data would contribute to the climatology which depends on averages (Line

numbers 215-222). The reason to worry about this balance is the following: if one cruise has HRMS data, the number of data can be in the thousands for that campaign. However, ten other historical cruises would have much lower number of data if they are using the GC technique. Hence, the final average, if computed with this raw data, would be biased heavily towards only one cruise, rather than being a climatological representative average. This is why we decided to degrade the temporal resolution to be more uniform. As HRMS data become more common, the bias towards one single campaign in a region will reduce and hence the climatology will not get biased towards a single study.

*Is it possible to demonstrate how these decision impacts the final climatology by repeating what was done for testing the sensitivity of results to using dynamic vs. static boundaries for provinces and VLS?*

**Response**: We calculated the difference observed due to temporal averaging of data into minutely, hourly, or daily bins as suggested. The annual global average was 1.86 for the minute binning, ~1.9 nM for the hourly binning and 1.96 nM for the daily binning. This is not a significant difference for the global average. However, the largest differences are observed where the HRMS sampling took place (off the coastal region of Antarctica) where the regional mean is biased towards single cruise averages for the minute averaging. We have now included this in the manuscript (line 237-240).

*I didn't understand parts of section 2.5. It is not clear how the 1degx1deg product was created and how it was merged with the province averages-based product? Are you doing a 1x1 bin averaging of the underlying raw data? It is stated that the province-based averages and the binned data are superimposed to merge them. I suspect some kind of weighted averaging is done; this needs to be explained.*

**Response**: In the first-guess fields, the pixels where the underlying raw data is available, are substituted with the averaged values for the particular pixels and then a Barnes filter was applied. We have clarified this in the manuscript now.

*In the Barnes filter step, the ROI is not explained (in one sentence the average distance between a grid point and the data points that influence it can be mentioned), but the sensitivity of results to this parameter is discussed using patchiness as a measure of accuracy. This is problematic because the climatology itself should be the best representation of reality, yet this makes it sound like you already know what reality should look like.*

**Response:** We have now included the definition of ROI in the manuscript (line 325). The patchiness in the DMS is one of the main issues that was highlighted in the last climatologies because such patchiness was not observed in the field. The reason for this was also recognised to the ROI in the Barnes filter. The fact that this patchiness reduces when we use field-observation generated ROI shows that the premise to reduce the patchiness was justified.

*VLS, which is actually based on data although it is not clear from which reference(s) the underlying VLS data are taken, is used as ROI in the Barnes filter. I'm not an expert in this kind of thing, but this made sense to me. However, I struggled to reconcile short VLS values with the starting point of this climatology, which is the notion that DMS concentration should more or less be the same within the same biogeochemical provinces. Perhaps a sentence or two are needed to bring everything together here.*

**Response:** The similarity of the biogeochemical parameters like chlorophyll, SST, bathymetry etc., define a biogeochemical province. The DMS values in contrast are affected by the higher variability within each province. A VLS value is essentially saying, how much can really high-resolution data be smoothed before it no longer captures most of that really high-resolution variability. This high-resolution variability reflects things like eddies. Using the example of eddies, a province's biogeochemistry could be defined by being an eddy-dominated regime which leads to a characteristic average biogeochemical field. The variability length scale and the biogeochemical province size would therefore be of very different magnitudes. We have now included more details regarding the VLS calculations in the manuscript in addition to a modified figure S5.

*A final suggestion: It would be useful if the new DMS flux estimates are used to calculate DMS emissions. The emissions could be presented in tabular form for each province and for 30deg latitude bands, and also as a graph vs. latitude, which could include the L11-based emissions for comparison. This would be helpful for atmospheric scientists studying atmospheric oxidation products of DMS.*

**Response**: We have prepared a table (Table 2) providing DMS emissions in 10º latitude bands to maintain the continuity with Lana et al (2011) along with the flux values the reported by Lana et al (2011) for comparison. A figure showing the difference between the fluxes (figure S16) is also added in the supplementary text.

*The specific comments below are listed in the order they appear in the paper, not in the order of importance.*

*Specific comments:*

*Ln 31: Missing full stop.*
**Response**: Corrected.

*Ln 91-93: I don't understand the sentence. Delete "incorporating" perhaps?*
**Response**: The sentence is reworded to "These approaches provided statistical relationships needed to understand the mechanisms of the biogeochemical cycle of DMS, its formation and removal from the surface ocean." (Line numbers 90-92)

*Ln 94: No need for comma.*
**Response**: Corrected.

*Ln 109: Which database?*
**Response**: Corrected.

*LN 139-141: It is stated here that data from highly productive regions lead to patchiness in L11. These sentences, and some others later, seem to imply DMS emissions from blooms should be ignored because the climatology should not look patchy. I agree that patchiness is not good when introduced during data analysis, for example due to introduction of provinces and related assumptions to come up with first guess estimates od DMS concentrations. However, if there is patchiness inherent to the underlying data, don't we need to accept that as reality?*
**Response**: We agree that if the patchiness is in the underlying data, it should be preserved. Using a 99.9 percentile filter removes not just anomalous blooms but also data which would be erroneous due to instrumental issues. Using the new algorithms, the 'real' patchiness is preserved. For example, the Rev3 climatology shows some patches in regions like south of Alaskan coast during northern summer months (Section 3.1.2) or off Chilean coast during southern summer months (Section 3.1.4) (Figure 6) but removes the patchiness introduced by data analysis.

*Ln 178: It is not clear what is meant by data should be within a range of 25%.*

**Response**: Rephrased to make this clear.

*Ln 195: I think the 0.001% is a typo. Should be 0.1% for 99.9% threshold.*
**Response**: This was a typo, corrected.

*Ln 209-210: Should be less than or equal to 24 hours?*
**Response**: It is greater than or equal to 24 hours because it is the sampling 'frequency'.

*Ln 211: Fig. 2a does not show any data collected less frequently than 24 hours. If lower frequencies are bundled into the 24 hour bin, this should be noted in the caption.*
**Response**: The data with frequency of 24 hours and more was binned together. We have changed the caption to mention this.

*Ln 214: Fig. 2b shows high-res campaigns are 15% of the total.*
**Response**: Corrected.

*Ln 215: No need for "which is based on observations."*
**Response**: Corrected.

*Ln 218-219: I don't understand what is meant by "matching?" Is 1 hour chosen to have roughly equal contribution from CIMS/MIMS vs. GC data?*
**Response**: Yes, to avoid biases. This is now corrected.

*Ln 235: "Computing a mean value across the provinces" is confusing to me, I think you mean calculating a mean value that is representative of the entire province?*
**Response**: We reworded the sentence to "Computing mean values representing these provinces …". (Line number 238)

*Ln 280: I think this is the first instance a province acronym being used. It would be easier to read the manuscript if the number associated with the provinces accompanied the acronyms every time an acronym is used because the maps only show the numbers.*
**Response**: We found it to be confusing to read, especially later when the concentrations are discussed. Hence, we decided on using the acronym and added the column of province numbers in the Table 1.

*Ln 300-304: I suggest seasonal cycle or month-to-month variability instead of annual trend.*

**Response**: We reworded the sentence using '*month-to-month variability*' in the appropriate places.

*Ln 350: Is there really data from under the sea-ice, or is this a result of calculating average values for provinces?*

**Response**: It is mostly due to the average values for provinces. The data could be from the time and region that wasn't ice-covered during the time of measurement but falls under 50% coverage with respect to the sea-ice climatology.

*Ln 373-375: Would the use of wind data from more recent periods change the fluxes appreciably? My guess is no. A one-to-one comparison to L11 argument is not that convincing because many aspects of the methods have been changed.*

**Response**:

We wanted to observe the effect of the new climatology on flux calculation. Hence, we kept the wind and SST climatology and the flux parameterisation constant and used Rev3 output instead of L11 output for making a valid one-to-one comparison.

*Ln 418-419: Do you mean southwest coast of Africa and the west coast of South America?*

**Response**:

The sentence is reworded to '…the southwest coast of Africa and the west coast of South America…' (Line number 420).

*Ln 441: You switched what number is reported in the parenthesis, I think.*

**Response**:

We checked this and found that the code returns the value with sea ice as higher than without sea ice. So, it is reported as is.

*Ln 583: Fig. S11b color bar label is not in percentage.*

**Response**:

The necessary correction is made.

*Ln 640: Should be Fig. S13.*

**Response**:

The necessary correction is made.

*Ln 646-648: Fig. S14 shows L11 annual average is slightly less than Rev3. Obviously, this depends on what is subtracted from what to make those figures. Caption should include this information.*

**Response**:

The necessary correction is made.